# MESSAGE PASSING NEURAL PDE SOLVERS

**Johannes Brandstetter**[*]
University of Amsterdam
Johannes Kepler University Linz
brandstetter@ml.jku.at

**Daniel E. Worrall**[*]
Qualcomm AI Research[†]
dworrall@qti.qualcomm.com

**Max Welling**
University of Amsterdam
m.welling@uva.nl

## ABSTRACT

The numerical solution of partial differential equations (PDEs) is difficult, having led to a century of research so far. Recently, there have been pushes to build neural–numerical hybrid solvers, which piggy-backs the modern trend towards fully end-to-end learned systems. Most works so far can only generalize over a subset of properties to which a generic solver would be faced, including: resolution, topology, geometry, boundary conditions, domain discretization regularity, dimensionality, etc. In this work, we build a solver, satisfying these properties, where all the components are based on neural message passing, replacing all heuristically designed components in the computation graph with backprop-optimized neural function approximators. We show that neural message passing solvers representationally contain some classical methods, such as finite differences, finite volumes, and WENO schemes. In order to encourage stability in training autoregressive models, we put forward a method that is based on the principle of zero-stability, posing stability as a domain adaptation problem. We validate our method on various fluid-like flow problems, demonstrating fast, stable, and accurate performance across different domain topologies, discretization, etc. in 1D and 2D. Our model outperforms state-of-the-art numerical solvers in the low resolution regime in terms of speed and accuracy.

## 1 INTRODUCTION

In the sciences, years of work have yielded extremely detailed mathematical models of physical phenomena. Many of these models are expressed naturally in differential equation form (Olver, 2014), most of the time as temporal partial differential equations (PDE). Solving these differential equations is of huge importance for problems in all numerate disciplines such as weather forecasting (Lynch, 2008), astronomical simulations (Courant et al., 1967), molecular modeling (Lelièvre & Stoltz, 2016) , or jet engine design (Athanasopoulos et al., 2009). Solving most equations of importance is analytically intractable and necessitates falling back on numerical approximation schemes. Obtaining accurate solutions of bounded error with minimal computational overhead requires the need for handcrafted solvers, always tailored to the equation at hand (Hairer et al., 1993).

The design of "good" PDE solvers is no mean feat. The perfect solver should satisfy an almost endless list of conditions. There are *user requirements*, such as being fast, using minimal computational overhead, being accurate, providing uncertainty estimates, generalizing across PDEs, and being easy to use. Then there are *structural requirements* of the problem, such as spatial resolution and timescale, domain sampling regularity, domain topology and geometry, boundary conditions, dimensionality, and solution space smoothness. And then there are *implementational requirements*, such as maintaining stability over long rollouts and preserving invariants. It is precisely because of this considerable list of requirements that the field of numerical methods is a *splitter* field (Bartels, 2016), tending to build handcrafted solvers for each sub-problem, rather than a *lumper* field, where a mentality of "one method to rule them all" reigns. This tendency is commonly justified with

---

[*]Equal contribution
[†]Qualcomm AI Research is an initiative of Qualcomm Technologies, Inc.

reference to *no free lunch theorems*. We propose to numerically solve PDEs with an end-to-end, neural solver. Our contributions can be broken down into three main parts: (i) An end-to-end fully neural PDE solver, based on neural message passing, which offers flexibility to satisfy all structural requirements of a typical PDE problem. This design is motivated by the insight that some classical solvers (finite differences, finite volumes, and WENO scheme) can be posed as special cases of message passing. (ii) *Temporal bundling* and the *pushforward trick*, which are methods to encourage *zero-stability* in training autoregressive models. (iii) Generalization across multiple PDEs within a given class. At test time, new PDE coefficients can be input to the solver.

## 2 BACKGROUND AND RELATED WORK

Here in Section 2.1 we briefly outline definitions and notation. We then outline some classical solving techniques in Section 2.2. Lastly, in Section 2.3, we list some recent neural solvers and split them into the two main neural solving paradigms for temporal PDEs.

### 2.1 PARTIAL DIFFERENTIAL EQUATIONS

We focus on PDEs in one time dimension $t = [0, T]$ and possibly multiple spatial dimensions $\mathbf{x} = [x_1, x_2, \ldots, x_D]^\top \in \mathbb{X}$. These can be written down in the form

$$\partial_t \mathbf{u} = F(t, \mathbf{x}, \mathbf{u}, \partial_{\mathbf{x}} \mathbf{u}, \partial_{\mathbf{xx}} \mathbf{u}, \ldots) \qquad (t, \mathbf{x}) \in [0, T] \times \mathbb{X} \qquad (1)$$

$$\mathbf{u}(0, \mathbf{x}) = \mathbf{u}^0(\mathbf{x}), \qquad B[\mathbf{u}](t, x) = 0 \qquad \mathbf{x} \in \mathbb{X}, \ (t, \mathbf{x}) \in [0, T] \times \partial \mathbb{X} \qquad (2)$$

where $\mathbf{u} : [0, T] \times \mathbb{X} \to \mathbb{R}^n$ is the *solution*, with *initial condition* $\mathbf{u}^0(\mathbf{x})$ at time $t = 0$ and *boundary conditions* $B[\mathbf{u}](t, x) = 0$ when $\mathbf{x}$ is on the boundary $\partial \mathbb{X}$ of the domain $\mathbb{X}$. The notation $\partial_{\mathbf{x}} \mathbf{u}, \partial_{\mathbf{xx}} \mathbf{u}, \ldots$ is shorthand for partial derivatives $\partial \mathbf{u} / \partial \mathbf{x}, \partial^2 \mathbf{u} / \partial \mathbf{x}^2$, and so forth. Most notably, $\partial \mathbf{u} / \partial \mathbf{x}$ represents a $n \times D$ dimensional Jacobian matrix, where each row is the transpose of the gradient of the corresponding component of $\mathbf{u}$. We consider *Dirichlet boundary conditions*, where the boundary operator $B_{\mathcal{D}}[\mathbf{u}] = \mathbf{u} - \mathbf{b}_{\mathcal{D}}$ for fixed function $\mathbf{b}_{\mathcal{D}}$ and *Neumann boundary conditions*, where $B_{\mathcal{N}}[u] = \mathbf{n}^\top \partial_{\mathbf{x}} u - b_{\mathcal{N}}$ for scalar-valued $u$, where $\mathbf{n}$ is an outward facing normal on $\partial \mathbb{X}$.

**Conservation form**  Among all PDEs, we hone in on solving those that can be written down in *conservation form*, because there is already precedent in the field for having studied these (Bar-Sinai et al., 2019; Li et al., 2020a). Conservation form PDEs are written as

$$\partial_t \mathbf{u} + \nabla \cdot \mathbf{J}(\mathbf{u}) = 0 , \qquad (3)$$

where $\nabla \cdot \mathbf{J}$ is the divergence of $\mathbf{J}$. The quantity $\mathbf{J} : \mathbb{R}^n \to \mathbb{R}^n$ is the *flux*, which has the interpretation of a quantity that appears to flow. Consequently, $\mathbf{u}$ is a conserved quantity within a volume, only changing through the net flux $\mathbf{J}(\mathbf{u})$ through its boundaries.

### 2.2 CLASSICAL SOLVERS

**Grids and cells**  Numerical solvers partition $\mathbb{X}$ into a finite *grid* $X = \{c_i\}_{i=1}^N$ of $N$ small non-overlapping volumes called *cells* $c_i \subset \mathbb{X}$. In this work, we focus on grids of rectangular cells. Each cell has a center at $\mathbf{x}_i$. $\mathbf{u}_i^k$ is used to denote the discretized solution in cell $c_i$ and time $t_k$. There are two main ways to compute $\mathbf{u}_i^k$: sampling $\mathbf{u}_i^k = \mathbf{u}(t_k, \mathbf{x}_i)$ and averaging $\mathbf{u}_i^k = \int_{c_i} \mathbf{u}(t_k, \mathbf{x}) \, \mathrm{d}\mathbf{x}$. In our notation, omitting an index implies that we use the entire *slice*, so $\mathbf{u}^k = (\mathbf{u}_1^k, \mathbf{u}_2^k, \ldots, \mathbf{u}_N^k)$.

**Method of lines**  A common technique to solve temporal PDEs is the *method of lines* (Schiesser, 2012), discretizing domain $\mathbb{X}$ and solution $\mathbf{u}$ into a grid $X$ and a vector $\mathbf{u}^k$. We then solve $\partial_t \mathbf{u}^t \big|_{t_k} = f(t, \mathbf{u}^k)$ for $t \in [0, T]$ , where $f$ is the form of $F$ acting on the vectorized $\mathbf{u}^t$ instead of the function $\mathbf{u}(t, \mathbf{x})$. The only derivative operator is now in time, making it an ordinary differential equation (ODE), which can be solved with off-the-shelf ODE solvers (Butcher, 1987; Everhart, 1985). $f$ can be formed by approximating spatial derivatives on the grid. Below are three classical techniques.

**Finite difference method (FDM)**  In FDM, spatial derivative operators (e.g., $\partial_{\mathbf{x}}$) are replaced with difference operators, called *stencils*. For instance, $\partial_x u^k \big|_{x_i}$ might become $(u_{i+1}^k - u_i^k)/(x_{i+1} - x_i)$. Principled ways to derive stencils can be found in Appendix A. FDM is simple and efficient, but suffers poor stability unless the spatial and temporal discretizations are carefully controlled.

**Finite volume method (FVM)**   FVM works for equations in conservation form. It can be shown via the divergence theorem that the integral of $\mathbf{u}$ over cell $i$ increases only by the net flux into the cell. In 1D, this leads to $f(t, u_i^k) = \frac{1}{\Delta x_i}(J_{i-1/2}^k - J_{i+1/2}^k)$ , where $\Delta x_i$ is the cell width, and $J_{i-1/2}^k, J_{i+1/2}^k$ the flux at the left and right cell boundary at time $t_k$, respectively. The problem thus boils down to estimating the flux at cell boundaries $x_{i\pm 1/2}$. The beauty of this technique is that the integral of $u$ is exactly conserved. FVM is generally more stable and accurate than FDM, but can only be applied to conservation form equations.

**Pseudospectral method (PSM)**   PSM computes derivatives in Fourier space. In practical terms, the $m^{\text{th}}$ derivative is computed as $\text{IFFT}\{(\iota\omega)^m \text{FFT}(u)\}$ for $\iota = \sqrt{-1}$. These derivatives obtain exponential accuracy (Tadmor, 1986), for smooth solutions on periodic domains and regular grids. For non-periodic domains, analogues using other polynomial transforms exist, but for non-smooth solution this technique cannot be applied.

## 2.3   Neural solvers

We build on recent exciting developments in the field to learn PDE solvers. These *neural PDE solvers*, as we refer to them, are laying the foundations of what is becoming both a rapidly growing and impactful area of research. Neural PDE solvers for temporal PDEs fall into two broad categories, *autoregressive methods* and *neural operator methods*, see Figure 1a.

**Neural operator methods**   Neural operator methods treat the mapping from initial conditions to solutions at time $t$ as an input–output mapping learnable via supervised learning. For a given PDE and given initial conditions $\mathbf{u}_0$, a neural operator $\mathcal{M} : [0, T] \times \mathcal{F} \to \mathcal{F}$, where $\mathcal{F}$ is a (possibly infinite-dimensional) function space, is trained to satisfy

$$\mathcal{M}(t, \mathbf{u}^0) = \mathbf{u}(t) . \tag{4}$$

Finite-dimensional operator methods (Raissi, 2018; Sirignano & Spiliopoulos, 2018; Bhatnagar et al., 2019; Guo et al., 2016; Zhu & Zabaras, 2018; Khoo et al., 2020), where $\dim(\mathcal{F}) < \infty$ are grid-dependent, so cannot generalize over geometry and sampling. Infinite-dimensional operator methods (Li et al., 2020c;a; Bhattacharya et al., 2021; Patel et al., 2021) by contrast resolve this issue. Each network is trained on example solutions of the equation of interest and is therefore locked to that equation. These models are not designed to generalize to dynamics for out-of-distribution $t$.

**Autogressive methods**   An orthogonal approach, which we take, is autoregressive methods. These solve the PDE iteratively. For time-independent PDEs, the solution at time $t + \Delta t$ is computed as

$$\mathbf{u}(t + \Delta t) = \mathcal{A}(\Delta t, \mathbf{u}(t)) , \tag{5}$$

where $\mathcal{A} : \mathbb{R}_{>0} \times \mathbb{R}^N \to \mathbb{R}^N$ is the temporal update. In this work, since $\Delta t$ is fixed, we just write $\mathcal{A}(\mathbf{u}(t))$. Three important works in this area are Bar-Sinai et al. (2019), Greenfeld et al. (2019), and Hsieh et al. (2019). Each paper focuses on a different class of PDE solver: finite volumes, multigrid, and iterative finite elements, respectively. Crucially, they all use a hybrid approach (Garcia Satorras et al., 2019), where the solver computational graph is preserved and heuristically-chosen parameters are predicted with a neural network. Hsieh et al. (2019) even have convergence guarantees for their method, something rare in deep learning. Hybrid methods are desirable for sharing structure with classical solvers. So far in the literature, however, it appears that autoregressive methods are more the exception than the norm, and for those methods published, it is reported that they are hard to train. In Section 3 we explore why this is and seek to remedy it.

## 3   Method

In this section we detail our method in two parts: training framework and architecture. The training framework tackles the *distribution shift problem* in autoregressive solvers, which leads to instability. We then outline the network architecture, which is a message passing neural network.

### 3.1   Training framework

Autoregressive solvers map solutions $\mathbf{u}^k$ to causally consequent ones $\mathbf{u}^{k+1}$. A straightforward way of training is *one-step training*. If $p_0(\mathbf{u}^0)$ is the distribution of initial conditions in the training set, and $p_k(\mathbf{u}^k) = \int p(\mathbf{u}^k|\mathbf{u}^0)p_0(\mathbf{u}^0)\,\mathrm{d}\mathbf{u}^0$ is the groundtruth distribution at iteration $k$, we minimize

$$L_{\text{one-step}} = \mathbb{E}_k \mathbb{E}_{\mathbf{u}^{k+1}|\mathbf{u}^k, \mathbf{u}^k \sim p_k} \left[ \mathcal{L}(\mathcal{A}(\mathbf{u}^k), \mathbf{u}^{k+1}) \right] , \tag{6}$$

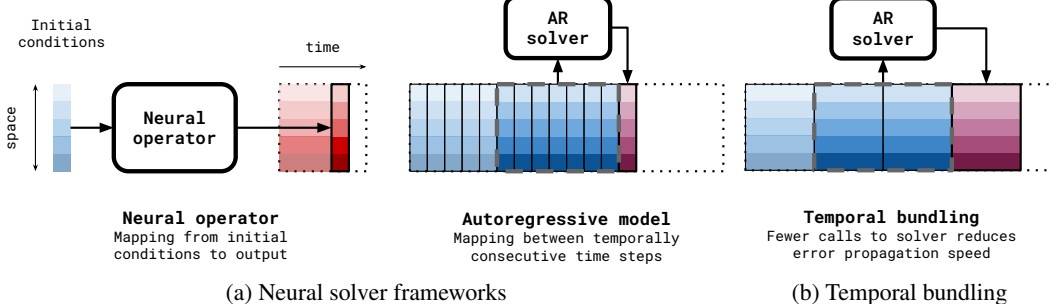

(a) Neural solver frameworks        (b) Temporal bundling

Figure 1: (a) LEFT: Neural operators perform a direct mapping from initial conditions to a solution at time $t$. RIGHT: Autoregressive models on the other hand compute the solution at time $t$ based on the computed solution at a fixed time offset before. (b) Our autoregressive solver outputs multiple time slices on every call, to reduce number of solver calls and therefore error propagation speed.

where $\mathcal{L}$ is an appropriate loss function. At test time, this method has a key failure mode, instability: small errors in $\mathcal{A}$ accumulate over rollouts greater in length than 1 (which is the vast majority of rollouts), and lead to divergence from the groundtruth. This can be interpreted as overfitting to the one-step training distribution, and thus being prone to generalize poorly if the input shifts from this, which is usually the case after a few rollout steps.

**The pushforward trick** We approach the problem in probabilistic terms. The solver maps $p_k \mapsto \mathcal{A}_\sharp p_k$ at iteration $k+1$, where $\mathcal{A}_\sharp : \mathbb{P}(X) \to \mathbb{P}(X)$ is the pushforward operator for $\mathcal{A}$ and $\mathbb{P}(X)$ is the space of distributions on $X$. After a single test time iteration, the solver sees samples from $\mathcal{A}_\sharp p_k$ instead of the distribution $p_{k+1}$, and unfortunately $\mathcal{A}_\sharp p_k \neq p_{k+1}$ because errors always survive training. The test time distribution is thus shifted, which we refer to as the *distribution shift problem*. This is a domain adaptation problem. We mitigate the distribution shift problem by adding a stability loss term, accounting for the distribution shift. A natural candidate is an adversarial-style loss

$$L_{\text{stability}} = \mathbb{E}_k \mathbb{E}_{\mathbf{u}^{k+1}|\mathbf{u}^k, \mathbf{u}^k \sim p_k} \left[ \mathbb{E}_{\boldsymbol{\epsilon}|\mathbf{u}^k} \left[ \mathcal{L}(\mathcal{A}(\mathbf{u}^k + \boldsymbol{\epsilon}), \mathbf{u}^{k+1})\right]\right], \tag{7}$$

where $\boldsymbol{\epsilon}|\mathbf{u}^k$ is an adversarial perturbation sampled from an appropriate distribution. For the perturbation distribution, we choose $\boldsymbol{\epsilon}$ such that $(\mathbf{u}^k + \boldsymbol{\epsilon}) \sim \mathcal{A}_\sharp p_k$. This can be easily achieved by using $(\mathbf{u}^k + \boldsymbol{\epsilon}) = \mathcal{A}(\mathbf{u}^{k-1})$ for $\mathbf{u}^{k-1}$ one step causally preceding $\mathbf{u}^k$. Our total loss is then $L_{\text{one-step}} + L_{\text{stability}}$. We call this the *pushforward trick*. We implement this by unrolling the solver for 2 steps but only backpropagating errors on the last unroll step, as shown in Figure 2. This is also outlined algorithmically in the appendix. We found it important not to backpropagate through the first unroll step. This is not only faster, it also seems to be more stable. Exactly why, we are not sure, but we think it may be to ensure the perturbations are large enough. Training the adversarial distribution itself to minimize the error, defeats the purpose of using it as an adversarial distribution. Adversarial losses were also introduced in Sanchez-Gonzalez et al. (2020) and later used in Mayr et al. (2021), where Brownian motion noise is used for $\boldsymbol{\epsilon}$ and there is some similarity to Noisy Nodes (Godwin et al.), where noise injection is found to stabilize training of deep graph neural networks. There are also connections with *zero-stability* (Hairer et al., 1993) from the ODE solver literature. Zero-stability is the condition that perturbations in the input conditions are damped out sublinearly in time, that is $\|\mathcal{A}(\mathbf{u}^0 + \boldsymbol{\epsilon}) - \mathbf{u}^1\| < \kappa \|\boldsymbol{\epsilon}\|$, for appropriate norm and small $\kappa$. The pushforward trick can be seen to minimize $\kappa$ directly.

**The temporal bundling trick** The second trick we found to be effective for stability and reducing rollout time is to predict multiple timesteps into the future synchronously. A typical temporal solver only predicts $\mathbf{u}^0 \mapsto \mathbf{u}^1$; whereas, we predict $K$ steps $\mathbf{u}^0 \mapsto (\mathbf{u}^1, \mathbf{u}^2, ..., \mathbf{u}^K) = \mathbf{u}^{1:K}$ together. This reduces the number of solver calls by a factor of $K$ and so reduces the number of times the solution distribution undergoes distribution shifts. A schematic of this setup can be seen in Figure 1b.

## 3.2 ARCHITECTURE

We model the grid $X$ as a graph $\mathcal{G} = (\mathcal{V}, \mathcal{E})$ with nodes $i \in \mathcal{V}$, edges $ij \in \mathcal{E}$, and node features $\mathbf{f}_i \in \mathbb{R}^c$. The nodes represent grid cells $c_i$ and the edges define local neighborhoods. Modeling the domain as a graph offers flexibility over grid sampling regularity, spatial/temporal resolution,

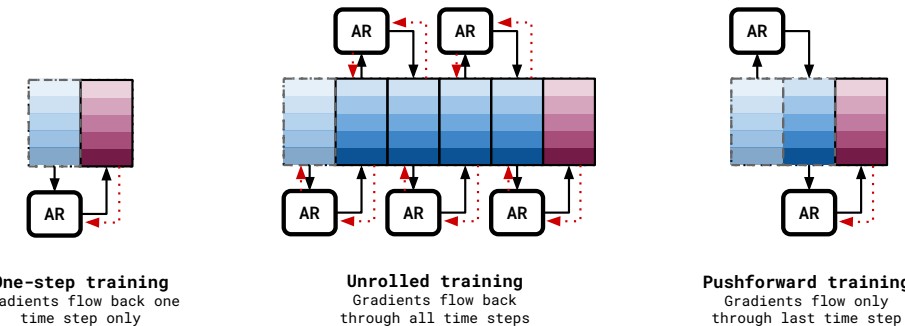

Figure 2: Different training strategies. LEFT: One-step training only predicts solutions one step into the future. MIDDLE: Unrolled training predicts $N$ steps into the future. RIGHT: Adversarial training predicts $N$ steps into the future, but only backprops on the last step.

domain topology and geometry, boundary modeling and dimensionality. The solver is a graph neural network (GNN) (Scarselli et al., 2009; Kipf & Welling, 2017; Defferrard et al., 2016; Gilmer et al., 2017; Battaglia et al., 2018), representationally containing the function class of several classical solvers, see Section 3.2. We follow the Encode-Process-Decode framework of Battaglia et al. (2018) and Sanchez-Gonzalez et al. (2020) , with adjustments. We are not the first to use GNNs as PDE solvers (Li et al., 2020b; De Avila Belbute-Peres et al., 2020), but ours have several notable features. Different aspects of the chosen architecture are ablated in Appendix G.

**Encoder** The encoder computes node embeddings. For each node $i$ it maps the last $K$ solution values $\mathbf{u}_i^{k-K:k}$, node position $\mathbf{x}_i$, current time $t_k$, and equation embedding $\boldsymbol{\theta}_{\text{PDE}}$ to node embedding vector $\mathbf{f}_i^0 = \epsilon^v([\mathbf{u}_i^{k-K:k}, \mathbf{x}_i, t_k, \boldsymbol{\theta}_{\text{PDE}}])$. $\boldsymbol{\theta}_{\text{PDE}}$ contains the PDE coefficients and other attributes such as boundary conditions. An exact description is found in Section 4. The inclusion of $\boldsymbol{\theta}_{\text{PDE}}$ allows us to train the solver on multiple different PDEs.

**Processor** The processor computes $M$ steps of learned message passing, with intermediate graph representations $\mathcal{G}^1, \mathcal{G}^2, ..., \mathcal{G}^M$. The specific updates we use are

$$\text{edge } j \to i \text{ message:} \qquad \mathbf{m}_{ij}^m = \phi\left(\mathbf{f}_i^m, \mathbf{f}_j^m, \mathbf{u}_i^{k-K:k} - \mathbf{u}_j^{k-K:k}, \mathbf{x}_i - \mathbf{x}_j, \boldsymbol{\theta}_{\text{PDE}}\right) , \qquad (8)$$

$$\text{node } i \text{ update:} \qquad \mathbf{f}_i^{m+1} = \psi\left(\mathbf{f}_i^m, \sum_{j \in \mathcal{N}(i)} \mathbf{m}_{ij}^m, \boldsymbol{\theta}_{\text{PDE}}\right) , \qquad (9)$$

where $\mathcal{N}(i)$ holds the neighbors of node $i$, and $\phi$ and $\psi$ are multilayer perceptrons (MLPs). Using relative positions $\mathbf{x}_j - \mathbf{x}_i$ can be justified by the translational symmetry of the PDEs we consider. Solution differences $\mathbf{u}_i - \mathbf{u}_j$ make sense by thinking of the message passing as a local difference operator, like a numerical derivative operator. Parameters $\boldsymbol{\theta}_{\text{PDE}}$ are inserted into the message passing similar to Brandstetter et al. (2021)

**Decoder** After message passing, we use a shallow 1D convolutional network with shared weights across spatial locations to output the $K$ next timestep predictions at grid point $\mathbf{x}_i$. For each node $i$, the processor outputs a vector $\mathbf{f}_i^M$. We treat this vector as a temporally contiguous signal, which we feed into a CNN over time. The CNN helps to smooth the signal over time and is reminiscent of *linear multistep methods* (Butcher, 1987), which are very efficient but generally not used because of stability concerns. We seem to have avoided these stability issues, by making the time solver nonlinear and adaptive to its input. The result is a new vector $\mathbf{d}_i = (\mathbf{d}_i^1, \mathbf{d}_i^2, ..., \mathbf{d}_i^K)$ with each element $\mathbf{d}_i^k$ corresponding to a different point in time. We use this to update the solution as

$$\mathbf{u}_i^{k+\ell} = \mathbf{u}_i^k + (t_{k+\ell} - t_k)\mathbf{d}_i^\ell , \qquad 1 \le \ell \le K . \qquad (10)$$

The motivation for this choice of decoder has to do with a property called *consistency* (Arnold, 2015), which states that $\lim_{\Delta t \to 0} \|\mathcal{A}(\Delta t, \mathbf{u}^0) - \mathbf{u}(\Delta t)\| = 0$, i.e. the prediction matches the exact solution in the infinitesimal time limit. Consistency is a requirement for zero-stability of the rollouts.

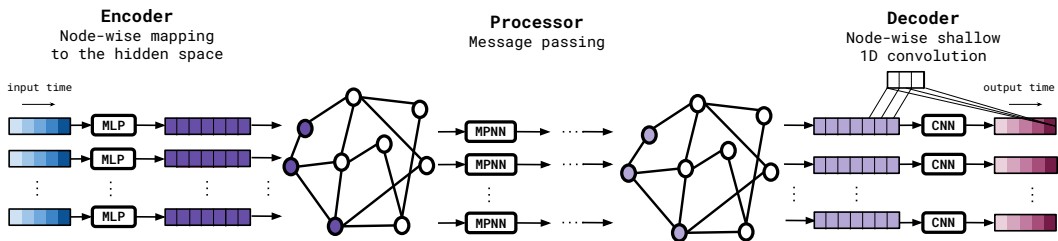

Figure 3: Schematic sketch of our MP-PDE Solver.

**Connections.** As mentioned in Bar-Sinai et al. (2019), both FDM and FVM are linear methods, which estimate $n^{\text{th}}$-order point-wise function derivatives as

$$[\partial_x^{(n)} u]_i \simeq \sum_{j \in \mathcal{N}(i)} \alpha_j^{(n)} u_j \tag{11}$$

for appropriately chosen coefficients $\alpha_j^{(n)}$, where $\mathcal{N}(i)$ is the neighborhood of cell $i$. FDM computes this at cell centers, and FVM computes this at cell boundaries. These estimates are plugged into flux equations, see Table 3 in the appendix, followed by an optional FVM update step, to compute time derivative estimates for the ODE solver. The WENO5 scheme computes derivative estimates by taking an adaptively-weighted average over multiple FVM estimates, computed using different neighborhoods of cell $i$ (see Equation 23). The FVM update, Equation 11, and Equation 23 are just message passing schemes with weighted aggregation (1 layer for FDM, 2 layers for FVM, and 3 layers for WENO). It is through this connection, that we see that message-passing neural networks representationally contain these classical schemes, and are thus a well-motivated architecture.

## 4 EXPERIMENTS

We demonstrate the effectiveness of the MP-PDE solver on tasks of varying difficulty to showcase its qualities. In 1D, we study its ability to generalize to unseen equations within a given family; we study boundary handling for periodic, Dirichlet, and Neumann boundary conditions; we study both regular and irregular grids; and we study the ability to model shock waves. We then show that the MP-PDE is able to solve equations in 2D. We also run ablations over the pushforward trick and variations, to demonstrate its utility. As baselines, we compare against standard classical PDE solvers, namely; FDM, pseudospectral methods, and a WENO5 solver, and we compare against the Fourier Neural Operator of Li et al. (2020a) as an example of a state-of-the-art neural operator method. The MP-PDE solver architecture is detailed in Appendix F.

### 4.1 INTERPOLATING BETWEEN PDES

**Data** We focus on the family of PDEs

$$[\partial_t u + \partial_x(\alpha u^2 - \beta \partial_x u + \gamma \partial_{xx} u)](t, x) = \delta(t, x), \tag{12}$$

$$u(0, x) = \delta(0, x), \qquad \delta(t, x) = \sum_{j=1}^{J} A_j \sin(\omega_j t + 2\pi \ell_j x / L + \phi_j). \tag{13}$$

Writing $\boldsymbol{\theta}_{\text{PDE}} = (\alpha, \beta, \gamma)$, corner cases are the heat equation $\boldsymbol{\theta}_{\text{PDE}} = (0, \eta, 0)$, Burgers' equation $\boldsymbol{\theta}_{\text{PDE}} = (0.5, \eta, 0)$, and the KdV equation $\boldsymbol{\theta}_{\text{PDE}} = (3, 0, 1)$. The term $\delta$ is a forcing term, following Bar-Sinai et al. (2019), with $J = 5$, $L = 16$ and coefficients sampled uniformly in $A_j \in [-0.5, 0.5], \omega_j \in [-0.4, -0.4], \ell_j \in \{1, 2, 3\}, \phi_j \in [0, 2\pi)$. This setup guarantees periodicity of the initial conditions and forcing. Space is uniformly discretized to $n_x = 200$ cells in $[0, 16)$ with periodic boundary and time is uniformly discretized to $n_t = 200$ points in $[0, 4]$. Our training sets consist of 2096 trajectories, downsampled to resolutions $(n_t, n_x) \in \{(250, 100), (250, 50), (250, 40)\}$. Numerical groundtruth is generated using a $5^{\text{th}}$-order WENO scheme (WENO5) (Shu, 2003) for the convection term $\partial_x u^2$ and $4^{\text{th}}$-order finite difference stencils for the remaining terms. The temporal solver is an explicit Runge-Kutta 4 solver (Runge, 1895; Kutta, 1901) with adaptive timestepping. Detailed methods and implementation are in Appendix C. All methods are implemented for GPU, so runtime comparisons are fair. For the interested reader comparison of WENO and FDM schemes against analytical solutions can be found in Appendix C to establish utility in generating groundtruth.

**Experiments and results**  We consider three scenarios: **E1** Burgers' equation without diffusion $\boldsymbol{\theta}_{\text{PDE}} = (1, 0, 0)$ for shock modeling; **E2** Burgers' equation with variable diffusion $\boldsymbol{\theta}_{\text{PDE}} = (1, \eta, 0)$ where $0 \leq \eta \leq 0.2$; and **E3** a mixed scenario with $\boldsymbol{\theta}_{\text{PDE}} = (\alpha, \beta, \gamma)$ where $0.0 \leq \alpha \leq 3.0$, $0.0 \leq \beta \leq 0.4$ and $0.0 \leq \gamma \leq 1.0$. **E2** and **E3** test the generalization capability. We compare against downsampled groundtruth (WENO5) and a variation of the Fourier Neural Operator with an autoregressive structure (FNO-RNN) used in Section 5.3 of their paper, and trained with unrolled training (see Figure 2). For our models we run the MP-PDE solver, an ablated version (MP-PDE-$\boldsymbol{\theta}_{\text{PDE}}$), without $\boldsymbol{\theta}_{\text{PDE}}$ features, and the Fourier Neural Operator method trained using our temporal bundling and pushforward tricks (FNO-PF). Errors and runtimes for all experiments are in Table 1. We see that the MP-PDE solver outperforms WENO5 and FNO-RNN in accuracy. Temporal bundling and the pushforward trick improve FNO dramatically, to the point where it beats MP-PDE on **E1**. But MP-PDE outperforms FNO-PF on **E2** and **E3** indicating that FNO is best for single equation modeling, but MP-PDE is better at generalization. MP-PDE predictions are best if equation parameters $\boldsymbol{\theta}_{\text{PDE}}$ are used, evidenced in **E2** and **E3**. This effect is most pronounced for **E3**, where all parameters are varied. Exemplary rollout plots for **E2** and **E3** are in Appendix F.1. Figure 4 (TOP) shows shock formation at different resolutions (**E1**), a traditionally difficult phenomenon to model—FDM and PSM methods cannot model shocks. Strikingly, shocks are preserved even at very low resolution.

## 4.2 VALIDATING TEMPORAL BUNDLING AND THE PUSHFORWARD METHOD

We observe solver survival times on **E1**, defined as the time until the solution diverges from groundtruth. A solution $\hat{\mathbf{u}}_i^k$ diverges from the groundtruth $\mathbf{u}_i^k$ when its max-normalized $L_1$-error $\frac{1}{n_x} \sum_{i=1}^{n_x} \frac{|\hat{\mathbf{u}}_i^k - \mathbf{u}_i^k|}{\max_j |\mathbf{u}_j^k|}$ exceeds 0.1. The solvers are unrolled to $n_t = 1000$ timesteps with $T = 16$ s. Examples are shown in Figure 4 (BOTTOM), where we observe increasing divergence after $\sim 8$ s. This is corroborated by Figure 5a where we see survival ratio against timestep. This is in line with observed problems with autoregressive models from the literature—see Figure C.3 of Sanchez-Gonzalez et al. (2020) or Figure S9 of Bar-Sinai et al. (2019)

In a second experiment, we compare the efficacy of the pushforward trick. Already, we saw that, coupled with temporal bundling, it improved FNO for our autoregressive tasks. In Figure 5b, we plot the survival ratios for models trained with and without the pushforward trick. As a third comparison we show a model trained with Gaussian noise adversarial perturbations, similar to that proposed in Sanchez-Gonzalez et al. (2020). We see that applying the pushforward trick leads to far higher survival times, confirming our model that instability can be addressed with adversarial training.

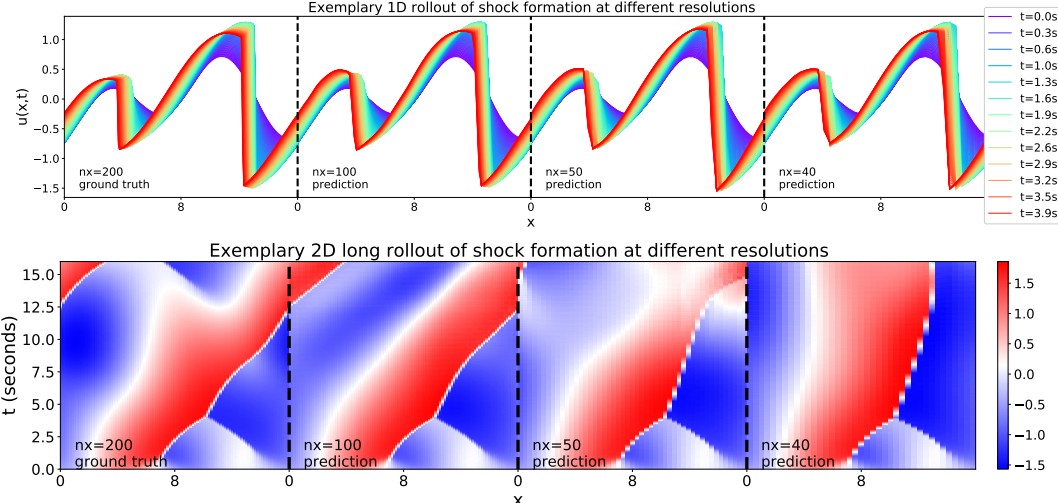

Figure 4: TOP: Exemplary 1D rollout of shock formation at different resolutions. The different colors represent PDE solutions at different timepoints. Both the small and the large shock are neatly captured and preserved even for low resolutions; boundary conditions are perfectly modeled. BOTTOM: Exemplary long 2D rollout of shock formations over 1000 timesteps. Different colors represent PDE solutions at different space-time points.

Table 1: Error and runtime experiments targeting shock wave formation modeling and generalization to unseen equations. Runtimes are for one full unrolling over 250 timesteps on a GeForce RTX 2080 Ti GPU. FNO-PF, MP-PDE-$\theta_{\text{PDE}}$, and MP-PDE are all ours. Accumulated error is $\frac{1}{n_x}\sum_{x,t}$ MSE.

| | $(n_t, n_x)$ | Accumulated Error ↓ | | | | | Runtime [s] ↓ | |
|---|---|---|---|---|---|---|---|---|
| | | WENO5 | FNO-RNN | FNO-PF | MP-PDE-$\theta_{\text{PDE}}$ | MP-PDE | WENO5 | MP-PDE |
| **E1** | $(250, 100)$ | 2.02 | 11.93 | **0.54** | - | 1.55 | 1.9 | 0.09 |
| **E1** | $(250, 50)$ | 6.23 | 29.98 | **0.51** | - | 1.67 | 1.8 | 0.08 |
| **E1** | $(250, 40)$ | 9.63 | 10.44 | **0.57** | - | 1.47 | 1.7 | 0.08 |
| **E2** | $(250, 100)$ | 1.19 | 17.09 | 2.53 | 1.62 | **1.58** | 1.9 | 0.09 |
| **E2** | $(250, 50)$ | 5.35 | 3.57 | 2.27 | 1.71 | **1.63** | 1.8 | 0.09 |
| **E2** | $(250, 40)$ | 8.05 | 3.26 | 2.38 | 1.49 | **1.45** | 1.7 | 0.08 |
| **E3** | $(250, 100)$ | 4.71 | 10.16 | 5.69 | 4.71 | **4.26** | 4.8 | 0.09 |
| **E3** | $(250, 50)$ | 11.71 | 14.49 | 5.39 | 10.90 | **3.74** | 4.5 | 0.09 |
| **E3** | $(250, 40)$ | 15.94 | 20.90 | 5.98 | 7.78 | **3.70** | 4.4 | 0.09 |

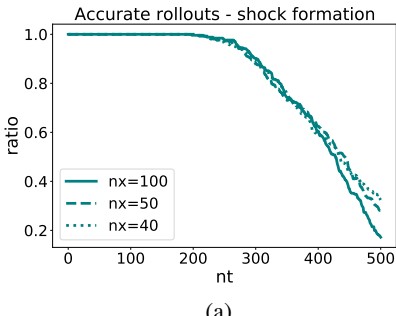

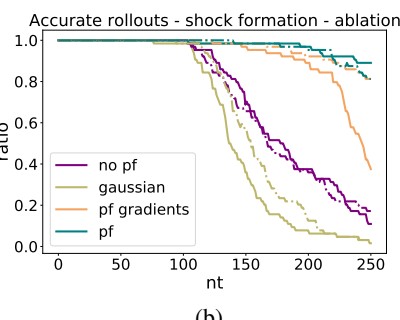

(a)                (b)

Figure 5: Survival times at E1. Rollout for long trajectories of 8 s (left), pushforward (pf) ablation (right). The ablation compares survival times at resolutions $n_x = 100$ (solid) and $n_x = 50$ (dashed) against survival times using pushforward (no pf), no pushforward but putting Gaussian noise ($\sigma = 0.01$), pushforward but without cutting the gradients (pf gradients).

Interestingly, injecting Gaussian perturbations appears worse than using none. Closer inspection of rollouts shows that although Gaussian perturbations improve stability, they lead to lower accuracy, by nature of injecting noise into the system.

### 4.3 SOLVING ON IRREGULAR GRIDS WITH DIFFERENT BOUNDARY CONDITIONS

The underlying motives for designing this experiment are to investigate (i) how well our MP-PDE solver can operate on irregular grids and (ii) how well our MP-PDE solver can generalize over different boundary conditions. Non-periodic domains and grid sampling irregularity go hand in hand, since pseudo-spectral methods designed for closed intervals operate on non-uniform grids.

**Data** We consider a simple 1D wave equation

$$\partial_{tt}u - c^2\partial_{xx}u = 0, \qquad x \in [-8, 8] \tag{14}$$

where $c$ is wave velocity ($c = 2$ in our experiments). We consider Dirichlet $B[u] = u = 0$ and Neumann $B[u] = \partial_x u = 0$ boundary conditions. This PDE is 2nd-order in time, but can be rewritten as 1st-order in time, by introducing the auxilliary variable $v = \partial_t u$ and writing $\partial_t[u, v] - [v, c^2\partial_{xx}u] = 0$. The initial condition is a Gaussian pulse with peak at random location. Numerical groundtruth is generated using FVM and Chebyshev spectral derivatives, integrated in time with an implicit Runge-Kutta method of Radau IIA family, order 5 (Hairer et al., 1993). We solve for groundtruth at resolution $(n_t, n_x) = (250, 200)$ on a Chebyshev extremal point grid (cell edges are located at $x_i = \cos(i\pi/(n_x + 1))$).

**Experiments and results** We consider three scenarios: **WE1** Wave equation with Dirichlet boundary conditions; **WE2** Wave equation with Neumann boundary conditions and, **WE3** Arbitrary combinations of these two boundary conditions, testing generalization capability of the MP-PDE solver.

Ablation studies (marked with $\theta_{\text{PDE}}$) have no equation specific parameters input to the MP-PDE solver. Table 2 compares our MP-PDE solver against state-of-the art numerical pseudospectral solvers. MP-PDE solvers obtain accurate results for low resolutions where pseudospectral solvers break. Interestingly, MP-PDE solvers can generalize over different boundary conditions, which gets more pronounced if boundary conditions are injected into the equation via $\theta_{\text{PDE}}$ features.

Table 2: Error and runtime comparison on tasks with non-periodic boundaries and irregular grids. Runtimes measure one full 250 timesteps unrolling on a GeForceRTX 2080 Ti GPU for MP-PDE solvers, and on a CPU for our pseudospectral (PS) solver implementation based on `scipy`.

| $(n_t, n_x)$ | $\frac{1}{n_x}\sum_{x,t}$MSE $\downarrow$ (**WE1**) | | $\frac{1}{n_x}\sum_{x,t}$MSE $\downarrow$ (**WE2**) | | $\frac{1}{n_x}\sum_{x,t}$MSE $\downarrow$ (**WE3**) | | | Runtime [s] $\downarrow$ | |
| --- | --- | --- | --- | --- | --- | --- | --- | --- | --- |
| | PS | MP-PDE | PS | MP-PDE | PS | MP-PDE $\theta_{\text{PDE}}$ | MP-PDE | PS | MP-PDE |
| (250, 100) | 0.004 | 0.137 | 0.004 | 0.111 | 0.004 | 38.775 | 0.097 | 0.60 | 0.09 |
| (250, 50) | 0.450 | 0.035 | 0.681 | 0.034 | 0.610 | 20.445 | 0.106 | 0.35 | 0.09 |
| (250, 40) | 194.622 | 0.042 | 217.300 | 0.003 | 204.298 | 16.859 | 0.219 | 0.25 | 0.09 |
| (250, 20) | breaks | 0.059 | breaks | 0.007 | breaks | 17.591 | 0.379 | 0.20 | 0.07 |

## 4.4 2D EXPERIMENTS

We finally test the scalability of our MP-PDE solver to a higher number of spatial dimensions, more specifically to 2D experiments. We use data from PHIFLOW[1], an open-source fluid simulation toolkit. We look at fluid simulation based on the Navier-Stokes equations, and simulate smoke inflow into a $32 \times 32$ grid, adding more smoke after every time step which follows the buoyancy force. Dynamics can be described by semi-Lagrangian advection for the velocity and MacCormack advection for the smoke distribution. Simulations run for 100 timesteps where one timestep corresponds to one second. Smoke inflow locations are sampled randomly. Architectural details are in Appendix F. Figure 12 in the appendix shows results of the MP-PDE solver and comparisons to the groundtruth simulation. The MP-PDE solver is able to capture the smoke inflow accurately over the given time period, suggesting scalability of MP-PDE solver to higher dimensions.

## 5 CONCLUSION

We have introduced a fully neural MP-PDE solver, which representationally contains classical methods, such as the FDM, FVM, and WENO schemes. We have diagnosed the distribution shift problem and introduced the pushforward trick combined with the idea of temporal bundling trying to alleviate it. We showed that these tricks reduce error explosion observed in training autoregressive models, including a SOTA neural operator method (Li et al., 2020a). We also demonstrated that MP-PDE solvers offer flexibility when generalizing across spatial resolution, timescale, domain sampling regularity, domain topology and geometry, boundary conditions, dimensionality, and solution space smoothness. In doing so, MP-PDE solvers are much faster than SOTA numerical solvers.

MP-PDE solvers cannot only be used to predict the solution of PDEs, but can e.g. also be re-interpreted to optimize the integration grid and the parameters of the PDE. For the former, simply position updates need to be included in the processor, similar to Satorras et al. (2021). For the latter, a trained MP-PDE solver can be fitted to new data where only $\theta_{\text{PDE}}$ features are adjusted. A limitation of our model is that we require high quality groundtruth data to train. Indeed generating this data in the first place was actually the toughest part of the whole project. However, this is a limitation of most neural PDE solvers in the literature. Another limitation is the lack of accuracy guarantees typical solvers have been designed to output. This is a common criticism of such learned numerical methods. A potential fix would be to fuse this work with that in probabilistic numerics (Hennig et al., 2015), as has been done for RK4 solvers (Schober et al., 2014). Another promising follow-up direction is to research alternative adversarial-style losses as introduced in Equation 7. Finally, we remark that leveraging symmetries and thus fostering generalization is a very active field of research, which is especially appealing for building neural PDE solvers since every PDE is defined via a unique set of symmetries (Olver, 1986).

---

[1]`https://github.com/tum-pbs/PhiFlow`

## 6 REPRODUCIBILITY STATEMENT

All data used in this work is generated by ourselves. It is thus of great importance to make sure that the produced datasets are correct. We therefore spend an extensive amount of time cross-checking our produced datasets. For experiments E1, E2, E3, this is done by comparing the WENO scheme to analytical solutions as discussed in detail in Appendix Section C.3, where we compare our implemented WENO scheme against two analytical solutions of the Burgers equation from literature. For experiments W1, W2, W3, we cross-checked if Gaussian wave packages keep their form throughout the whole wave propagation phase. Furthermore, the wave packages should change sign for Dirichlet boundary conditions and keep the sign for Neumann boundary conditions. Examples can be found in Appendix Section D.

We have described our architecture in Section 3.2 and provided further implementation details in Appendix Section F. We have introduced new concepts, namely temporal bundling and the pushforward method. We have described these concepts at length in our paper. We have validated temporal bundling and pushforward methods on both our and the Fourier Neural Operator (FNO) method Li et al. (2020a). We have not introduced new mathematical results. However, we have used data generation concepts from different mathematical fields and therefore have included a detailed description of those in our appendix.

For reproducibility, we provide our at https://github.com/brandstetter-johannes/MP-Neural-PDE-Solvers.

## 7 ETHICAL STATEMENT

The societal impact of MP-PDE solvers is difficult to predict. However, as stated in the introduction, solving differential equations is of huge importance for problems in many disciplines such as weather forecasting, astronomical simulations, or molecular modeling. As such, MP-PDE solvers potentially help to pave the way towards shortcuts for computationally expensive simulations. Most notably, a drastical computational shortcut is always somehow related to reducing the carbon footprint. However, in this regard, it is also important to remind ourselves that relying on simulations or now even shortcuts of those always requires monitoring and thorough quality checks.

### ACKNOWLEDGMENTS

Johannes Brandstetter thanks the Institute of Advanced Research in Artificial Intelligence (IARAI) and the Federal State Upper Austria for the support. The authors thank Markus Holzleitner for helpful comments on this work.

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

## A    INTERPOLATION

To compute classical numerical derivatives of a function $u : \mathbb{R} \to \mathbb{R}$, which has been sampled on a mesh of points $x_1 < x_2 < ... < x_N$ it is common to first fit a piecewise polynomial $p : \mathbb{R} \to \mathbb{R}$ on the mesh. We assume we have function evaluations $u_i = u(x_i)$ at the mesh nodes for all $i = 1, ..., N$ and once we have fitted the polynomial, we can use its derivatives at any new off-mesh point, for instance the half nodes $x_{i+\frac{1}{2}}$. Here we illustrate how to carry out this procedure.

A polynomial of degree $N-1$ (note the highest degree polynomial we can fit to $N$ points has degree $N-1$) can be fitted at the mesh points by solving the following linear system in $\mathbf{a}$

$$\underbrace{\begin{bmatrix} u_1 \\ \vdots \\ u_N \end{bmatrix}}_{\mathbf{u}} = \begin{bmatrix} p(x_1) \\ \vdots \\ p(x_N) \end{bmatrix} = \begin{bmatrix} \sum_{i=0}^{N-1} a_i x_1^i \\ \vdots \\ \sum_{i=0}^{N-1} a_i x_N^i \end{bmatrix} = \underbrace{\begin{bmatrix} x_1^0 & \cdots & x_1^{N-1} \\ \vdots & \ddots & \vdots \\ x_N^0 & \cdots & x_N^{N-1} \end{bmatrix}}_{\mathbf{X}} \underbrace{\begin{bmatrix} a_0 \\ \vdots \\ a_{N-1} \end{bmatrix}}_{\mathbf{a}}. \tag{15}$$

To find the polynomial interpolation at a new point $x$, we then do

$$p(x) = \mathbf{x}^\top \mathbf{a} = \mathbf{x}^\top \mathbf{X}^{-1} \mathbf{u} \tag{16}$$

where $\mathbf{x}^\top = [1, x, x^2, ..., x^{N-1}]$. To retrieve the $m^{\text{th}}$ derivative, where $m < N-1$, is also very simple. For this we have that

$$\frac{\mathrm{d}^m p}{\mathrm{d}x^m} = \sum_{i=0}^{N-1} a_i \frac{\mathrm{d}^m x^i}{\mathrm{d}x^m} = \sum_{i=0}^{N-1} a_i \cdot (i)_m \cdot x^{i-m} = \mathbf{x}^{(m)\top} \mathbf{a} = \mathbf{x}^{(m)\top} \mathbf{X}^{-1} \mathbf{u}, \tag{17}$$

$$\text{where } \mathbf{x}^{(m)\top} = [(0)_m x^{0-m}, (1)_m x^{1-m}, ..., (N-1)_m x^{N-1-m}] \tag{18}$$

where $(i)_m = i(i-1)(i-2) \cdots (i-m+1)$ is the $m^{\text{th}}$ Pochhammer symbol and $(i)_0 := 1$.

Note that is typical to fold $\mathbf{s} = \mathbf{x}^\top \mathbf{X}^{-1}$ into a single object, which we call a *stencil*.

## B    RECONSTRUCTION

Reconstruction is the task of fitting a piecewise polynomial on a mesh of points, when instead of functions values at the grid points we have cell averages $\bar{u}_i$, where

$$\bar{u}_i = \frac{1}{\Delta x_i} \int_{I_i} u(x) \, \mathrm{d}x, \tag{19}$$

where each cell $I_i = [x_{i-\frac{1}{2}}, x_{i+\frac{1}{2}}]$, the half nodes are defined as $x_{i+\frac{1}{2}} := \frac{1}{2}(x_i + x_{i+1})$ and $\Delta x_i = x_{i+\frac{1}{2}} - x_{i-\frac{1}{2}}$. The solution is to note that we can fit a polynomial $P(x)$ to the integral of $u(x)$, which will be exact at the half nodes, so

$$P(x_{i+\frac{1}{2}}) = U(x_{i+\frac{1}{2}}) = \int_{x_{1-\frac{1}{2}}}^{x_{i+\frac{1}{2}}} u(x) \, \mathrm{d}x = \sum_{k=1}^{i} \int_{x_{k-\frac{1}{2}}}^{x_{k+\frac{1}{2}}} u(x) \, \mathrm{d}x = \sum_{k=1}^{i} \bar{u}_k \Delta x_k. \tag{20}$$

We can then differentiate this polynomial to retrieve an estimate for the $u$ at off-mesh locations. Recall that polynomial differentiation is easy with the interpolant differentiation operators $\mathbf{x}^{(m)}$. The system of equations we need to solve is thus

$$\underbrace{\begin{bmatrix} \Delta x_1 & \cdots & 0 \\ \vdots & \ddots & \vdots \\ \Delta x_1 & \cdots & \Delta x_N \end{bmatrix}}_{\mathbf{L}} \underbrace{\begin{bmatrix} \bar{u}_1 \\ \vdots \\ \bar{u}_N \end{bmatrix}}_{\bar{\mathbf{u}}} = \underbrace{\begin{bmatrix} x_1^0 & \cdots & x_1^{N-1} \\ \vdots & \ddots & \vdots \\ x_N^0 & \cdots & x_N^{N-1} \end{bmatrix}}_{\mathbf{X}} \underbrace{\begin{bmatrix} a_0 \\ \vdots \\ a_{N-1} \end{bmatrix}}_{\mathbf{a}}. \tag{21}$$

where $\bar{\mathbf{u}}$ is the vector of cell averages and $\mathbf{L}$ is a lower triangular matrix performing the last sum in Equation 20. Thus the $m^{\text{th}}$ derivative of the polynomial $p(x) = P'(x)$ is

$$\frac{\mathrm{d}^m p}{\mathrm{d}x^m} = \frac{\mathrm{d}^{m+1} P}{\mathrm{d}x^{m+1}} = \mathbf{x}^{(m+1)\top} \mathbf{X}^{-1} \mathbf{L} \bar{\mathbf{u}} = \bar{\mathbf{s}}^{(m)\top} \bar{\mathbf{u}}. \tag{22}$$

## C  WENO Scheme

The *essentially non-oscillating* (ENO) scheme is an interpolation or reconstruction scheme to estimate function values and derivatives of a discontinuous function. The main idea is to use multiple overlapping stencils to estimate a derivative at point $x \in [x_i, x_{i+1}]$. We design $N$ stencils to fit the function on shifted overlapping intervals $I_1, ..., I_N$, where $I_k = [x_{i-N+1+k}, x_{i+k}]$. In the case of WENO reconstruction these intervals are $I_k = [x_{i-N+1+k-\frac{1}{2}}, x_{i+k+\frac{1}{2}}]$. If a discontinuity lies in $I = \bigcup_k I_k$, then it is likely that one of the substencils $\mathbf{s}_k^{(m)}$ or $\bar{\mathbf{s}}_k^{(m)}$ (defined on interval $I_k$) will not contain the discontinuity. We can thus use the substencil $\mathbf{s}_k^{(m)}$ or $\bar{\mathbf{s}}_k^{(m)}$ from the relatively smooth region to estimate the function derivatives at $x$. In the following, we focus on WENO reconstruction.

The *weighted essentially non-oscillating* (WENO) scheme goes one step further and takes a convex combination of the substencils to create a larger stencil $\bar{\mathbf{s}}$ on $I$, where (dropping the superscript for brevity)

$$\bar{\mathbf{s}} = \sum_{k=1}^{N} w_k \bar{\mathbf{s}}_k. \tag{23}$$

Here the *nonlinear weights* $w_k$ satisfy $\sum_{k=1}^{N} w_k = 1$. It was shown in Jiang & Shu (1996) that these nonlinear weights can be constructed as

$$w_k = \frac{\tilde{w}_k}{\sum_{j=1}^{N} \tilde{w}_j} \qquad \tilde{w}_k = \frac{\gamma_k}{(\epsilon + \beta_k)^2}, \tag{24}$$

where $\gamma_k$ is called the *linear weight*, $\epsilon$ is a small number, and $\beta_k$ is the *smoothness indicator*. The linear weights are set such that the sum $\sum_{k=1}^{N} \gamma_k \bar{\mathbf{s}}_k$ over the order $N-1$ stencils matches a larger order $2N-2$ stencil. The smoothness indicators are computed as

$$\beta_k = \sum_{m=1}^{N-1} \Delta x_k^{2m-1} \int_{x_{i-\frac{1}{2}}}^{x_{i+\frac{1}{2}}} \left( \frac{\mathrm{d}^m p_k}{\mathrm{d} x^m} \right)^2 \mathrm{d}x, \tag{25}$$

where $p_k$ is the polynomial corresponding to substencil $\bar{\mathbf{s}}_k$.

### C.1  WENO5 scheme

A conservative finite difference spatial discretization approximates a derivative $f(u)_x$ by a conservative difference

$$f(u)_x|_{x=x_i} \approx \frac{1}{\Delta x} \left( \hat{f}_{i+\frac{1}{2}} - \hat{f}_{i-\frac{1}{2}} \right) , \tag{26}$$

where $\hat{f}_{i+\frac{1}{2}}$ and $\hat{f}_{i-\frac{1}{2}}$ are numerical fluxes. Since $g(u)_y$ is approximated in the same way, finite difference methods have the same format for more than one spatial dimensions. The left-reconstructed (reconstruction is done from left to right) fifth order finite difference WENO scheme (WENO5) (Shu, 2003) has the $\hat{u}_{i+\frac{1}{2}}^{-}$ given by:

$$\hat{u}_{i+\frac{1}{2}}^{-} = w_1 \hat{u}_{i+\frac{1}{2}}^{-(1)} + w_2 \hat{u}_{i+\frac{1}{2}}^{-(2)} + w_3 \hat{u}_{i+\frac{1}{2}}^{-(3)} . \tag{27}$$

In equation 27, $\hat{u}_{i+\frac{1}{2}}^{-(j)}$ are the left WENO reconstructions on three different stencils given by

$$\hat{u}_{i+\frac{1}{2}}^{-(1)} = +\frac{1}{3} u_{i-2} - \frac{7}{6} u_{i-1} + \frac{11}{6} u_i , \tag{28}$$

$$\hat{u}_{i+\frac{1}{2}}^{-(2)} = -\frac{1}{6} u_{i-1} + \frac{5}{6} u_i + \frac{1}{3} u_{i+1} , \tag{29}$$

$$\hat{u}_{i+\frac{1}{2}}^{-(3)} = +\frac{1}{3} u_i + \frac{5}{6} u_{i+1} - \frac{1}{6} u_{i+2} , \tag{30}$$

and the non-linear weights $w_j$ given by

$$w_j = \frac{\tilde{w}_j}{\sum_{k=1}^{3} w_k}, \qquad \tilde{w}_k = \frac{\gamma_k}{(\epsilon + \beta_k)^2} , \tag{31}$$

with $\gamma_{\{1,2,3\}} = \{\frac{1}{10}, \frac{3}{5}, \frac{3}{10}\}$, and $\epsilon$ a tiny-valued parameter to avoid the denominator becoming 0. The left smoothness indicators $\beta_k^-$ are given by:

$$\beta_1^- = \frac{13}{12}\left(u_{i-2} - 2u_{i-1} + u_i\right)^2 + \frac{1}{4}\left(u_{i-2} - 4u_{i-1} + 3u_i\right)^2 , \tag{32}$$

$$\beta_2^- = \frac{13}{12}\left(u_{i-1} - 2u_i + u_{i+1}\right)^2 + \frac{1}{4}\left(u_{i-1} - u_{i+1}\right)^2 , \tag{33}$$

$$\beta_3^- = \frac{13}{12}\left(u_i - 2u_{i+1} + u_{i+2}\right)^2 + \frac{1}{4}\left(u_i - 4u_{i+1} + 3u_{i+2}\right)^2 . \tag{34}$$

$$\tag{35}$$

The right-reconstructed WENO5 scheme has the $\hat{u}_{i-\frac{1}{2}}^+$ given similarly to $\hat{u}_{i+\frac{1}{2}}^-$ but with all coefficients flipped since the reconstruction is done from the other side (from right to left). The right reconstruction on three different stencils are given by

$$\hat{u}_{i-\frac{1}{2}}^{+(1)} = +\frac{1}{3}u_{i+2} - \frac{7}{6}u_{i+1} + \frac{11}{6}u_i , \tag{36}$$

$$\hat{u}_{i-\frac{1}{2}}^{+(2)} = -\frac{1}{6}u_{i+1} + \frac{5}{6}u_i + \frac{1}{3}u_{i-1} , \tag{37}$$

$$\hat{u}_{i-\frac{1}{2}}^{+(3)} = +\frac{1}{3}u_i + \frac{5}{6}u_{i-1} - \frac{1}{6}u_{i-2} , \tag{38}$$

and the right smoothness indicators $\beta_k^+$ are given by

$$\beta_1^+ = \frac{13}{12}\left(u_{i+2} - 2u_{i+1} + u_i\right)^2 + \frac{1}{4}\left(u_{i+2} - 4u_{i+1} + 3u_i\right)^2 , \tag{39}$$

$$\beta_2^+ = \frac{13}{12}\left(u_{i+1} - 2u_i + u_{i-1}\right)^2 + \frac{1}{4}\left(u_{i+1} - u_{i-1}\right)^2 , \tag{40}$$

$$\beta_3^+ = \frac{13}{12}\left(u_i - 2u_{i-1} + u_{i-2}\right)^2 + \frac{1}{4}\left(u_i - 4u_{i-1} + 3u_{i-2}\right)^2 . \tag{41}$$

$$\tag{42}$$

Both $\hat{u}^-$ and $\hat{u}^+$ are needed for full flux reconstruction as explained in the next section.

## C.2 FLUX RECONSTRUCTION

We consider flux reconstruction via Godunov. For Godunov flux, $\hat{f}_{i+\frac{1}{2}} = \hat{f}(u_{i+\frac{1}{2}})$ is reconstructed from $\hat{u}_{i+\frac{1}{2}}^+$ and $\hat{u}_{i+\frac{1}{2}}^-$ via:

$$\hat{f}(u_{i+\frac{1}{2}}) = \begin{cases} \min\limits_{u_{i+\frac{1}{2}}^- \leq u \leq u_{i+\frac{1}{2}}^+} f(u), & \text{if } u_{i+\frac{1}{2}}^- \leq u_{i+\frac{1}{2}}^+ \\ \max\limits_{u_{i+\frac{1}{2}}^- \leq u \leq u_{i+\frac{1}{2}}^+} f(u), & \text{if } u_{i+\frac{1}{2}}^- > u_{i+\frac{1}{2}}^+ \end{cases} \tag{43}$$

## C.3 COMPARING WENO SCHEME TO ANALYTICAL SOLUTIONS

**First analytical case.** An analytical solvable case for Burgers equation arises for the boundary conditions

$$u(t, 0) = u(t, 2\pi) ,$$

and the initial conditions

$$u(0, x) = -2\nu \frac{\partial \phi / \partial x}{\phi} + 4 , \tag{44}$$

where

$$\phi = \exp\left(\frac{-x^2}{4\nu}\right) + \exp\left[\frac{-(x-2\pi)^2}{4\nu(t+1)}\right] \tag{45}$$

$$\frac{\partial\phi}{\partial x} = -\frac{2x}{4\nu}\exp\left(\frac{-x^2}{4\nu}\right) - \frac{2(x-2\pi)}{4\nu} + \exp\left[\frac{-(x-2\pi)^2}{4\nu}\right] \tag{46}$$

$$= -\frac{0.5x}{\nu}\exp\left(\frac{-x^2}{4\nu}\right) - \frac{0.5(x-2\pi)}{\nu}\exp\left[\frac{-(x-2\pi)^2}{4\nu}\right] . \tag{47}$$

The analytical solutions for this specific set of boundary and initial conditions gives

$$u(t,x) = -2\nu\frac{\partial\phi/\partial x}{\phi} + 4 , \tag{48}$$

where

$$\phi = \exp\left(\frac{-(x-4t)^2}{4\nu(t+1)}\right) + \exp\left[\frac{-(x-4t-2\pi)^2}{4\nu(t+1)}\right] \tag{49}$$

$$\frac{\partial\phi}{\partial x} = -\frac{2(x-4t)}{4\nu(t+1)}\exp\left(\frac{-(x-4t)^2}{4\nu(t+1)}\right) - \frac{2(x-4t-2\pi)}{4\nu(t+1)} + \exp\left[\frac{-(x-4t-2\pi)^2}{4\nu(t+1)}\right] \tag{50}$$

$$= -\frac{0.5(x-4t)}{\nu(t+1)}\exp\left(\frac{-(x-4t)^2}{4\nu(t+1)}\right) - \frac{0.5(x-4t-2\pi)}{\nu(t+1)}\exp\left[\frac{-(x-4t-2\pi)^2}{4\nu(t+1)}\right] . \tag{51}$$

For this first analytical solveable case, the analytical solution, the WENO scheme and the fourth order finite difference scheme (FDM) are compared in Fig. 6 for a diffusion term of $\nu = 0.005$. The WENO scheme models the analytical solution perfectly, whereas the FDM scheme fails to capture the shock accurately. For lower values of $\nu$ the effect gets even stronger.

**Second analytical case.** Another analytical solvable case for the Burgers equation arises for the boundary condition:

$$u(t,\pm 1) = 0 , \tag{52}$$

and the initial condition

$$u(0,x) = -\sin(\pi x) . \tag{53}$$

Solutions are (Basdevant et al., 1986)

$$u(t,x) = \frac{-\int_{-\infty}^{\infty}\sin\pi(x-\eta)f(x-\eta)\exp(-\eta^2/4\nu t)d\eta}{\int_{-\infty}^{\infty}f(x-\eta)\exp(-\eta^2/4\nu t)d\eta} , \tag{54}$$

with $f(y) = \exp(-\cos(\frac{\pi y}{2\pi\nu}))$ . Using Hermite integration allows the computation of accurate results up to $t = 3/\pi$.

For this second analytical solveable case, the analytical solution, and the WENO scheme are compared in Fig. 7 for a diffusion term of $\nu = 0.002$. The WENO scheme models the analytical solution perfectly. Modeling via the FDM scheme fails completely.

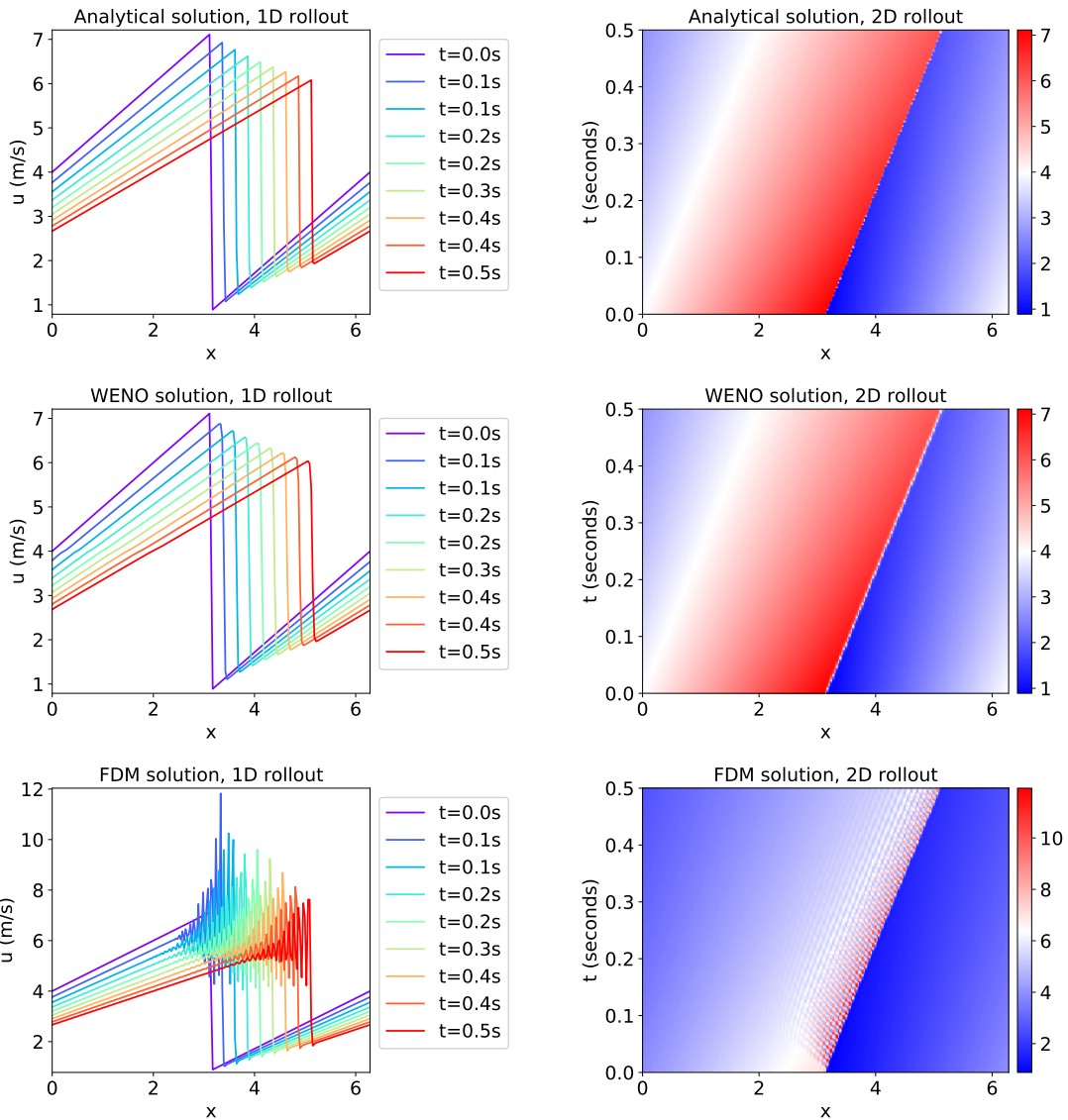

Figure 6: 1D and 2D rollouts for the first analytical case setting the diffusion term $\nu = 0.005$. Analytical solution (top), WENO scheme (middle) and Finite Difference scheme (FDM, bottom). The WENO scheme models the analytical solution perfectly, whereas the FDM scheme fails to capture the shock accurately.

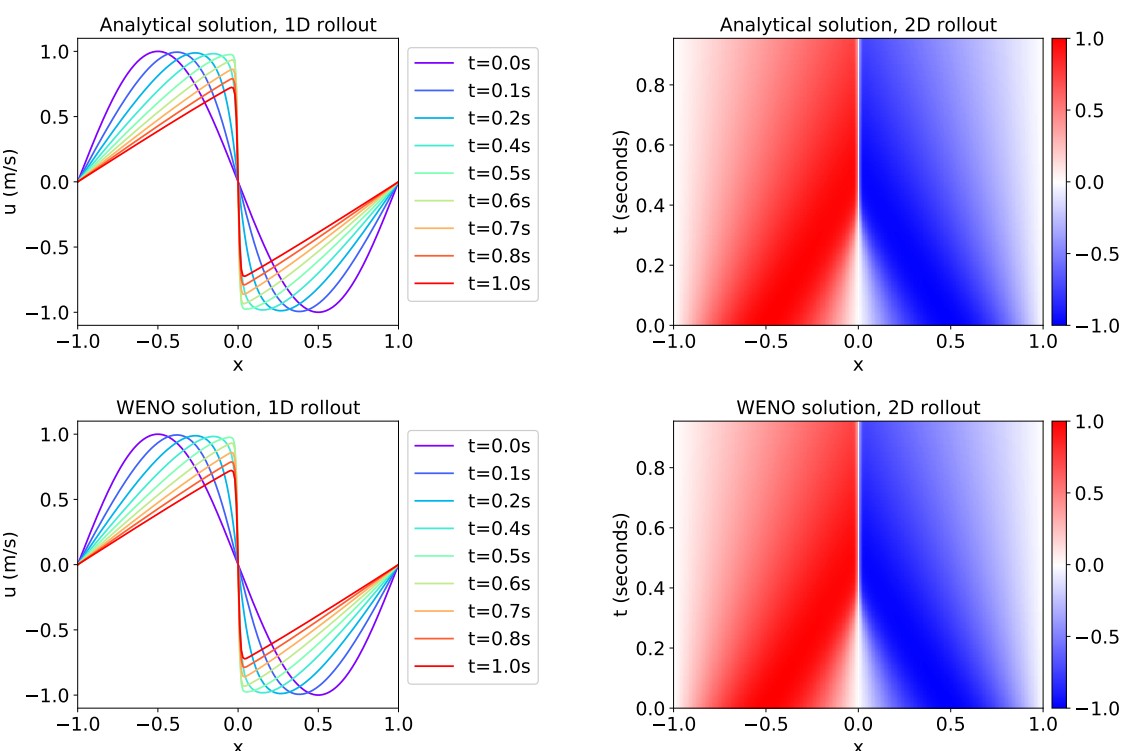

Figure 7: 1D and 2D rollouts for the second analytical case setting the diffusion term $\nu = 0.005$. Analytical solution (top), and WENO scheme solution(bottom).The WENO scheme models the analytical solution perfectly.

# D    PSEUDOSPECTRAL METHODS FOR WAVE PROPAGATION ON IRREGULAR GRIDS

We consider Dirichlet $B[u] = u = 0$ and Neumann $B[u] = \partial_x u = 0$ boundary conditions. Numerical groundtruth is generated using FVM and Chebyshev spectral derivatives, integrated in time with an implicit Runge-Kutta method of Radau IIA family, order 5 (Hairer et al., 1993). To properly fulfill the boundary conditions, wave packages have to travel between the boundaries and are bounced back with same and different sign for Neumann and Dirichlet boundary condition, respectively. Exemplary wave propagation for both boundary conditions is shown in Figure 8.

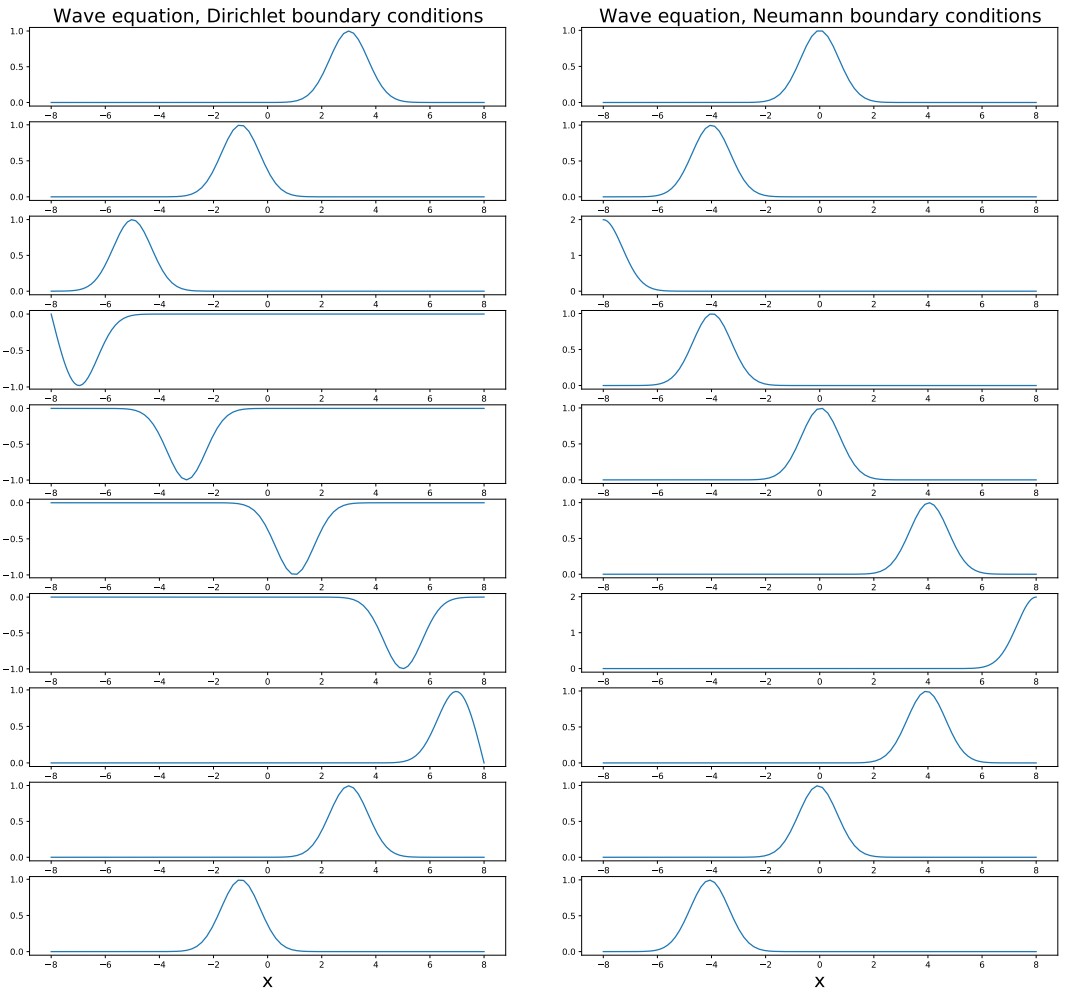

Figure 8: Exemplary wave propagation data for Dirichlet boundary conditions (left) and Neumann boundary conditions (right). Solutions are obtained on irregular grids using pseudospectral solvers.

# E    EXPLICIT RUNGE-KUTTA METHODS

The family of Runge-Kutta methods (Butcher, 1987) is given by:

$$u_{t_{n+1}} = u_{t_n} + \Delta t \sum_{i=1}^{s} b_i k_i \,, \tag{55}$$

where

$$k_1 = f(t_n, u_{t_n}) \,, \tag{56}$$
$$k_2 = f(t_n + c_2 \Delta t, u_{t_n} + h(a_{21} k_1)) \,, \tag{57}$$
$$k_3 = f(t_n + c_3 \Delta t, u_{t_n} + h(a_{31} k_1 + a_{32} k_2)) \,, \tag{58}$$
$$\vdots \tag{59}$$
$$k_s = f(t_n + c_s \Delta t, u_{t_n} + h(a_{s1} k_1 + a_{s2} k_2, \dots a_{s,s-1} k_{s-1})) \,. \tag{60}$$

For a particular Runge-Kutta method one needs to provide the number of stages $s$, and the coefficients $a_{ij}(1 \leq j < i \leq s)$, $b_i(i = 1, 2, \dots, s)$ and $c_i(i = 1, 2, \dots, s)$. These data are usually arranged in so-called Butcher tableaux (Butcher, 1963).

# F    EXPERIMENTS

Flux terms of equations that we study—the Heat, Burgers, Korteweg-de-Vries (KdV), and Kuromoto-Shivashinsky (KS) equation—are summarized in Table 3.

Table 3: 1D flux terms $J(u)$ of the Heat, Burgers, Korteweg-de-Vries (KdV), and Kuramoto-Shivashinsky (KS) equation.

|  | Heat | Burgers | KdV | KS |
|---|---|---|---|---|
| $J(u)$ | $-\eta\partial_x u$ | $\frac{1}{2}u^2 - \eta\partial_x u$ | $3u^2 + \partial_{xx}u$ | $\frac{1}{2}u^2 + \partial_x u + \partial_{xxx}u$ |

**Pushforward trick and temporal bundling.**    Pseudocode for one training step using the pushforward trick and temporal bundling is sketch in Algorithm 1.

---

**Algorithm 1** Pushforward trick and temporal bundling. For a given batched input data trajectory and a model, we draw a random timepoint $t$, get our input data trajectory, perform $N$ forward passes, and finally perform the supervised learning task with the according labels. $K$ is the number of steps we predict into the future using the temporal bundling trick, $N$ is number of unrolling steps in order to apply the pushforward trick, $T$ is the number of available timesteps in the training set.

---

**Require:** data, model, $N, K, T$                                              ▷ data is the complete PDE trajectory
   $t \leftarrow$ DrawRandomNumber $t \in \{1, ..., T\}$                     ▷ We draw a random starting point
   input $\leftarrow$ data$(t - K{:}t)$                                      ▷ We input the last $K$ timesteps
   **for** $n \in \{1, \ldots, N\}$ **do**
      input $\leftarrow$ model(input)
   **end for**                                                              ▷ Here we cut the gradients
   target $\leftarrow$ data$(t + NK{:}t + N(K + 1))$                ▷ Actual labels after the pushforward operation
   output $\leftarrow$ model(input)
   loss $\leftarrow$ criterion(output, target)

---

**Implementation details.**    MP-PDE architectures, consist of three parts (sequentially applied):

1. Encoder: Input $\rightarrow$ {fully-connected layer $\rightarrow$ activation $\rightarrow$ fully-connected layer $\rightarrow$ activation }, where fully connected layers are applied node-wise

2. Processor: 6 message passing layers as described in Sec. 3.2. Each layer consists of a 2-layer edge update network $\phi$ following Equation (8), and a 2-layer node update network $\psi$ following Equation (9).

3. Decoder: 1D convolutional network with shard weights across spatial locations $\rightarrow$ {1D CNN layer $\rightarrow$ activation $\rightarrow$ 1D CNN layer }

We optimize models using the AdamW optimizer (Loshchilov & Hutter, 2017) with learning rate 1e-4, weight decay 1e-8 for 20 epochs and minimize the root mean squared error (RMSE). We use batch size 16 for experiments **E1**-**E3** and **WE1**-**WE3** and batch size of 4 for 2D experiments. For experiments **E1**-**E3** we use a hidden size of 164, and for experiments **WE1**-**WE3** we use a hidden size of 128. In order to enforce zero-stability during training we unroll the solver for a maximum of 2 steps (see Sec. 3.1).

Message and update network in the processor consist of $\rightarrow$ {fully-connected layer $\rightarrow$ activation $\rightarrow$ fully-connected layer $\rightarrow$ activation }. We use skip-connections in the message passing layers and apply instance normalization (Ulyanov et al., 2016) for experiments **E1**-**E3** and **WE1**-**WE3**, and batch normalization (Ioffe & Szegedy, 2015) for the 2D experiments. For the decoder, we use 8 channels between the two CNN layers (1 input channel, 1 output channel) across all experiments. We use Swish (Ramachandran et al., 2017) activation functions for experiments **E1**-**E3** and **WE1**-**WE3** and ReLU activation for the 2D experiments. ReLU activation proved most effective for 2D experiments since a characteristic of the smoke inflow dynamics we studied is that values are zero for all positions which are untouched by smoke buoyancy at a given timepoint.

**Training details.**    The overall used architectures consist of roughly 1 million parameters and training for the different experiments takes between 12 and 24 hours on average on a GeForceRTX 2080

Ti GPU. 6 message passing layers and a hidden size of 128 for the 2-layer edge update network and the 2-layer node update network is a robust choice. A hidden size of 64 shows signs of underfitting, whereas a hidden size of 256 is not improve performance significantly. For the overall performance, more important than the number of parameters is the choice of the output 1D CNN, the choice of inputs to the edge update network $\phi$ following Equation (8) and the node update network $\psi$ following Equation (9). We ablate these choices in Appendix G.

Another interesting hyperparameter is the number of neighbors used for message passing. We construct our graphs by restricting the neighbors (edges) via a cutoff radius based on positional coordinates for experiments **E1**-**E3** and the 2D experiments. We effectively use 6 neighbors for experiments **E1**-**E3** and 8 neighbors for the 2D experiments. For experiments **WE1**-**WE3**, cutoff radii for selecting neighbors are not a robust choice since the grids are irregular and relative distances are much lower close to the boundaries. We therefore construct our graphs via a $k$-NN criterion, and effectively use between 20 neighbors (highest spatial resolution) and 6 neighbors (lowest spatial resolution).

### F.1    EXPERIMENTS E1, E2, E3

Figure 9 displays exemplary 1D rollouts at different resolution for the Burgers' equation with different diffusion terms. Lower diffusion coefficients result in faster shock formation. Figure 10 displays exemplary 1D rollouts for different parameter sets. Large $\alpha$ parameters result in fast and large shock formations. The wiggles arising due to the dispersive term $\gamma$ and cannot be captured by numerical solvers at low resolution. Our MP-PDE solver is able to capture these wiggles and reproduce them even at very low resolution.

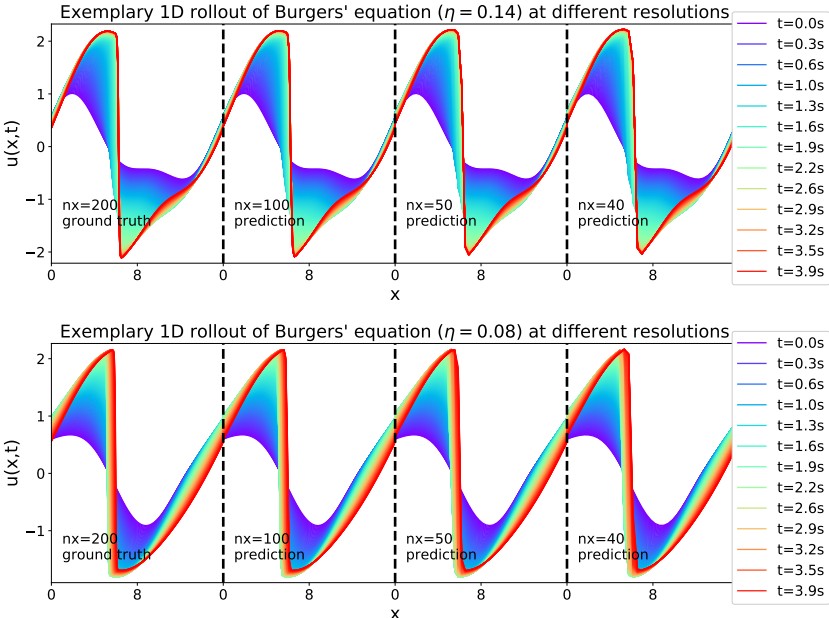

Figure 9: Exemplary 1D rollout of the Burgers' equation at different resolutions. The different colors represent PDE solutions at different timepoints. Diffusion coefficients of $\eta = 0.14$ (top) and $\eta = 0.08$ (bottom) are compared for the same initial conditions. Lower diffusion coefficients result in faster shock formation.

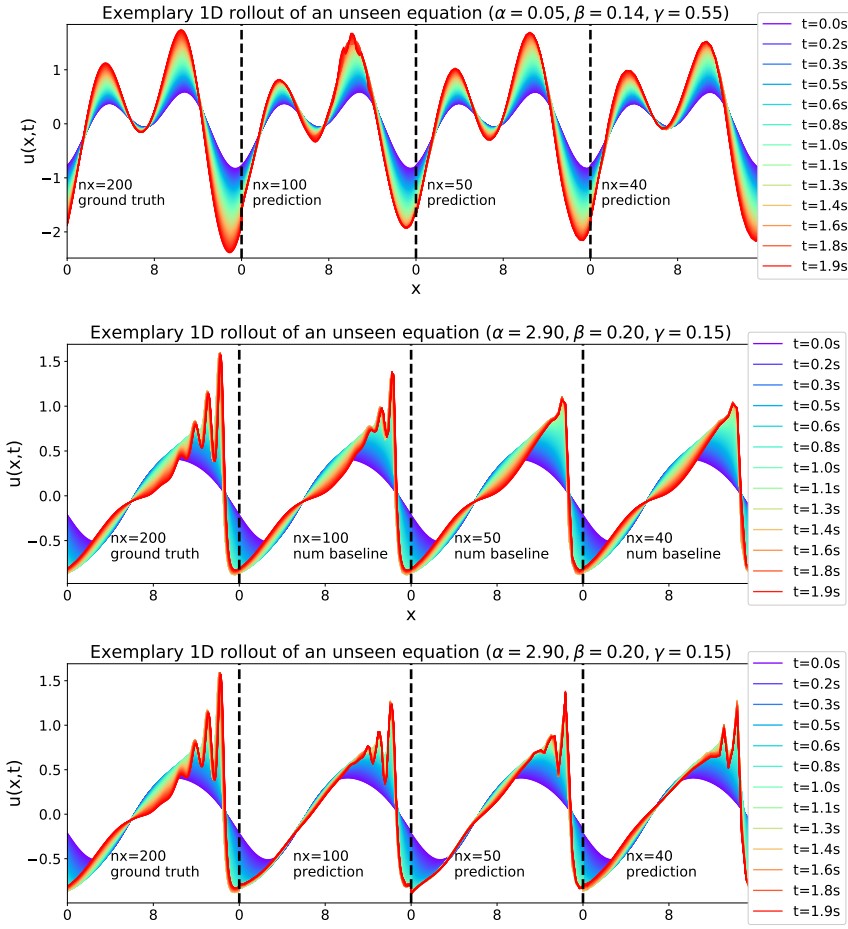

Figure 10: Exemplary 1D rollout an unseen equation with different equation parameters. The different colors represent PDE solutions at different timepoints. Low $\alpha$ parameters (top) result in diffusion like behavior. Large $\alpha$ parameters (middle, bottom) result in fast and large shock formations. The wiggles arising due to the dispersive term $\gamma$. Numerical solvers cannot capture the wiggles at low resolution (middle), MP-PDE solvers can reconstruct them much better (bottom).

## F.2   EXPERIMENTS WE1, WE2, WE3

Figure 11 displays exemplary 2D rollouts at different resolutions for the wave equation with Dirichlet and Neumann boundary conditions. Waves bounce back and forth between boundaries. MP-PDE solvers give accurate solutions on the irregular grids and are stable over time.

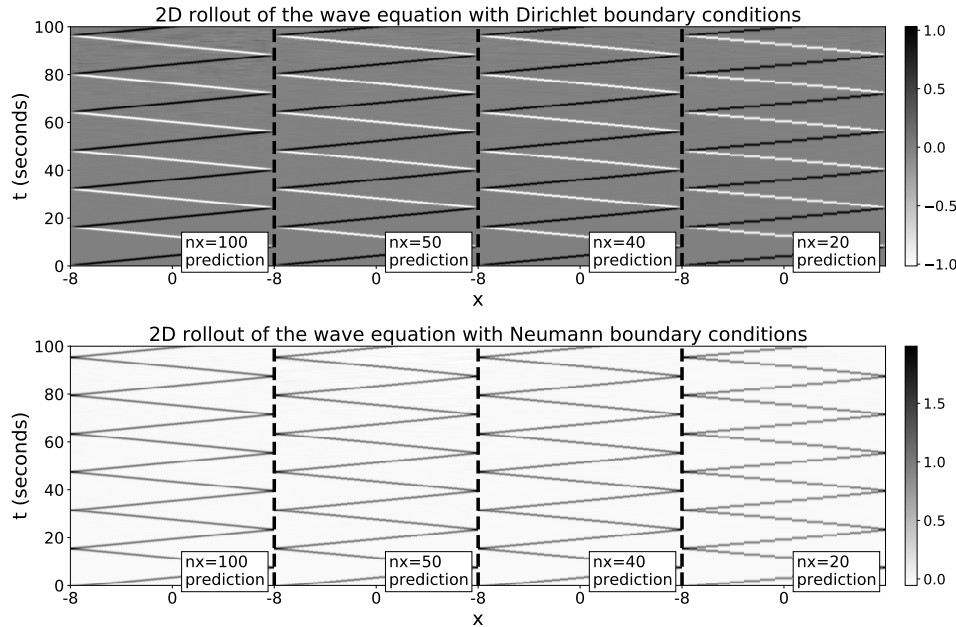

Figure 11: Exemplary 2D rollouts for the wave equation with Dirichlet boundary conditions (top) and Neumann boundary conditions (bottom). The different colors for the Dirichlet boundary condition comes from the fact that wave propagation changes the sign at each boundary.

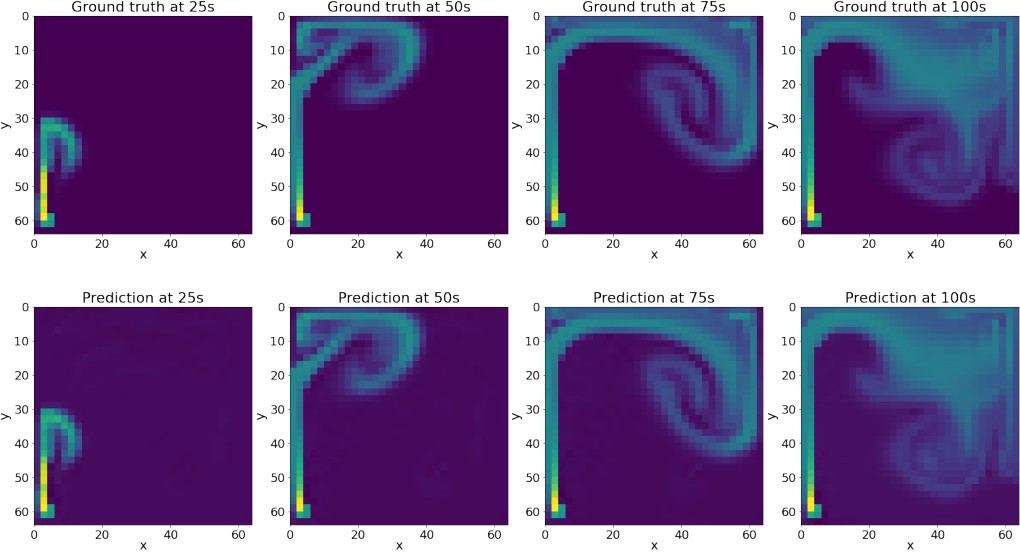

Figure 12: Exemplary 2D smoke inflow simulation. Ground truth data (top) are compared to MP-PDE solvers (bottom). Simulations run for 100 timesteps corresponding to 100 seconds. The MP-PDE solver is able to capture the smoke inflow acccurately over the given time period.

## G  ARCHITECTURE ABLATION AND COMPARISON TO CNNS

A schematic sketch of our MP-PDE solver is displayed in Figure 13 (sketch taken from the main paper). A GNN based architecture was chosen since GNNs have the potential to offer flexibility when generalizing across spatial resolution, timescale, domain sampling regularity, domain topology and geometry, boundary conditions, dimensionality, and solution space smoothness. The chosen architecture representationally contains classical methods, such as FDM, FVM, and WENO schemes. The architectures follows the Encode-Process-Decode framework of Battaglia et al. (2018), with adjustments. Most notably, PDE coefficients and other attributes such as boundary conditions denoted with $\boldsymbol{\theta}_{\mathrm{PDE}}$ are included in the processor. For the decoder, a shallow 2-layer 1D convolutional network with shared weights across spatial locations is applied, motivated by linear multistep methods (Butcher, 1987).

For ablating the architecture, three design choices are verified:

1. Does a GNN have the representational power of a vanilla convolutional network on a regular grid? We test against 1D and 2D baseline CNN architectures.

2. For the decoder part, how much does a 1D convolutional network with shared weights across spatial locations approve upon a standard MLP decoder?

3. How much does the inclusion of PDE coefficients $\boldsymbol{\theta}_{\mathrm{PDE}}$ help to generalize over e.g. different equations or different boundary conditions?

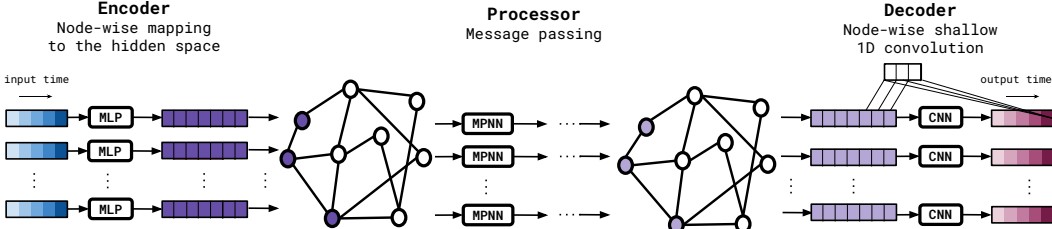

Figure 13: Schematic sketch of our MP-PDE Solver, sketch taken from the main paper.

Table 4 shows the three ablation (MP-PDE-$\boldsymbol{\theta}_{\mathrm{PDE}}$, MP-PDE-1D-CNN, Baseline CNN) tested on shock wave formation modeling and generalization to unseen experiments (experiments **E1**, **E2**, and **E3**), as described in Section 4.1 in the main paper. For the training of the different architectures the optimized training strategy consisting of temporal bundling and pushforward trick is used.

The **MP-PDE-$\boldsymbol{\theta}_{\mathrm{PDE}}$ ablation** results are the same as reported in the main paper. The effect gets more prominent if more equation specific parameters are available ($\boldsymbol{\theta}_{\mathrm{PDE}}$ features), as it is the case for experiment **E3**. We also refer the reader to the experiments presented in Table 2, where MP-PDE solvers are shown to be able to generalize over different boundary conditions, which gets much stronger pronounced if boundary conditions are injected into the equation via $\boldsymbol{\theta}_{\mathrm{PDE}}$ features.

The **MP-PDE-1D-CNN** ablation replaces the shallow 2-layer 1D convolutional network with in the decoder with a standard 2-layer MLP. Performance slightly degrades for the MLP decoder which is most likely due to the better temporal modeling introduced by the shared weights of the 1D CNN network.

A **Baseline 1D-CNN** is built up of 8 1D-CNN layers, where the input consists of the spatial resolution ($n_x$). The $K$ previous timesteps used for temporal bundling are treated as $K$ input channels. The output consequently predicts the next $K$ timesteps for the same spatial resolution ($K$ output channels). Using this format, again both temporal bundling and the pushforward trick can be effectively applied. The implemented 1D-CNN layers are:

• Input layer with $K$ input channel, 40 output channels, kernel of size 3.

• 3 layers with 40 input channels, 40 output channels, kernel of size 5.

• 3 layers with 40 input channels, 40 output channels, kernel of size 7.

• 1 output layer with 40 input channels, $K$ output channel, kernel of size 7.

Residual connections are used between the layers, and ELU (Clevert et al., 2016) non-linearities are applied. Circular padding is implemented to reflect the periodic boundary conditions. The CNN output is a new vector $\mathbf{d}_i = (\mathbf{d}_i^1, \mathbf{d}_i^2, ..., \mathbf{d}_i^K)$ with each element $\mathbf{d}_i^k$ corresponding to a different point in time. Analogously to the MP-PDE solver, we use the output to update the solution as

$$\mathbf{u}_i^{k+\ell} = \mathbf{u}_i^k + (t_{k+\ell} - t_k)\mathbf{d}_i^\ell , \qquad 1 \le \ell \le K , \tag{61}$$

where $K$ is the output (and input) dimension. This baseline 1D-CNN is conceptually very similar to our MP-PDE solver.

A **Baseline 2D-CNN** is built up of 6 2D-CNN layers, where the input consists of the spatial resolution ($n_x$) and the $K$ previous timesteps used for temporal bundling. The output consequently predicts the next $K$ timesteps for the same spatial resolution. Using this format, both temporal bundling and the pushforward trick can be effectively applied. The implemented 2D-CNN layers are:

- Input layer with 1 input channel, 16 output channels, $3 \times 3$ kernel.

- 4 intermediate layers with 16 input channels, 16 output channels, $5 \times 5$ kernel.

- 1 output layer with 16 input channels, 1 output channel, $7 \times 7$ kernel.

Residual connections are used between the layers, and ELU (Clevert et al., 2016) non-linearities are applied. For the spatial dimension, circular padding is implemented to reflect the periodic boundary conditions, for the temporal dimension zero padding is used. The CNN output is a new vector $\mathbf{d}_i = (\mathbf{d}_i^1, \mathbf{d}_i^2, ..., \mathbf{d}_i^K)$ with each element $\mathbf{d}_i^k$ corresponding to a different point in time. Analogously to the MP-PDE solver and analogously to the 1D-CNN, we use the output to update the solution as shown in Equation (61).

The 1D-CNN baseline which is conceptually very close to our MP-PDE solver performs much better than the 2D-CNN. However, we see already when looking at the results of **E3** that generalization for the 1D-CNN across different PDEs becomes harder.

Table 4: Ablation study comparing MP-PDE results on experiments on shock wave formation modeling and generaliziation to unseen equations (**E1**, **E3**, and **E3**) to an MP-PDE-$\theta_{\text{PDE}}$ ablation, an MP-PDE-1D-CNN ablation, a baseline 1D-CNN and a baseline 2D-CNN architecture. Runtimes are for one full unrolling over 250 timesteps on a GeForce RTX 2080 Ti GPU. Accumulated error is $\frac{1}{n_x} \sum_{x,t}$ MSE.

| | | Accumulated Error ↓ | | | | | Runtime [s] ↓ | | |
|---|---|---|---|---|---|---|---|---|---|
| | $(n_t, n_x)$ | MP-PDE | MP-PDE-$\theta_{\text{PDE}}$ | MP-PDE-1D-CNN | 1D-CNN | 2D-CNN | MP-PDE | 1D-CNN | 2D-CNN |
| **E1** | (250, 100) | 1.55 | - | 2.41 | 3.45 | 25.70 | 0.09 | 0.02 | 0.16 |
| **E1** | (250, 50) | 1.67 | - | 2.69 | 3.88 | 32.42 | 0.08 | 0.02 | 0.15 |
| **E1** | (250, 40) | 1.47 | - | 2.50 | 3.07 | 37.13 | 0.008 | 0.02 | 0.14 |
| **E2** | (250, 100) | 1.58 | 1.62 | 2.59 | 3.32 | 30.09 | 0.09 | 0.02 | 0.16 |
| **E2** | (250, 50) | 1.63 | 1.71 | 2.31 | 2.89 | 30.87 | 0.08 | 0.02 | 0.15 |
| **E2** | (250, 40) | 1.45 | 1.49 | 2.80 | 2.98 | 35.93 | 0.08 | 0.02 | 0.15 |
| **E3** | (250, 100) | 4.26 | 4.71 | 6.26 | 9.15 | 42.37 | 0.09 | 0.02 | 0.16 |
| **E3** | (250, 50) | 3.74 | 10.90 | 5.15 | 7.69 | 45.41 | 0.09 | 0.02 | 0.15 |
| **E3** | (250, 40) | 3.70 | 7.78 | 7.27 | 6.77 | 53.87 | 0.09 | 0.02 | 0.15 |

