# OpenReview forum: "Message Passing Neural PDE Solvers"
_ICLR.cc/2022/Conference — ICLR 2022 Spotlight_

### Official Review · Reviewer_wxXT · 2021-11-02

**Correctness:** 4
**Technical Novelty And Significance:** 4
**Empirical Novelty And Significance:** 4
**Recommendation:** 8
**Confidence:** 4

**Main Review:**

##########################################################################

Summary:

The paper proposes a framework to solve generic PDEs written in conservation form, in an autoregressive fashion. The grid domain is considered as a graph, whose nodes contain information about solutions to the PDE until a given times steps, node positions, time steps, and hyperparameters information about the PDE. The node’s labels are first embedded in a latent space. Then message passing is applied and the output is decoded to predict the next n time steps solutions together.
The network is trained with an additional loss which promotes zero stability of the solver, by constraining the network to be invariant to small perturbations in the input solutions of previous time steps.
The approach is validated on a parameterized family of 1d and 2d PDEs, showing accuracy and generalization results that are above the state of the art of neural solvers and classical solver schemes. In addition, the paper show promising results dealing with irregular domains with different types of boundary conditions.


##########################################################################

Reasons for score:



In my opinion, the paper brings significant advances to the field of neural PDE solvers. The proposed push forward trick, together with the temporal bundling trick, are valuable contributions as they impose favorably condition on autoregressive solvers (zero stability), improve the accuracy of other existing approaches as [Li et al 2020 a] and the inference time of the solver.
The approach is well evaluated experimentally,  showing accuracy and generalization results that are above the state of the art of neural solvers and beat classical solver schemes in inference time.
For these reasons I consider the manuscript to be accepted at ICLR 2022.




##########################################################################Pr

 Pros:

1. The paper tackles the important task of solving many families of PDEs, obtaining accuracy and generalization results upon the state-of-the-art of neural solvers.

2. The approach is well motivated as classical solver schemes (FDM, FVM, WENO) are instances of message passing, as the authors themselves stated.

3. The pushforward trick is by itself a valuable contribution as it enhances the performance of other existing neural solvers ([Li et al 2020 a]) as the authors show in Table 2.

4. The paper is clear and well written, providing sufficient background for people not experts in the field.


##########################################################################

Cons and questions:

1. It is not entirely clear to me how the network can generalize to multiple spatial resolutions: more specifically how the encoder and the message passing mechanism handle a varying number of nodes (i.e. cells) in the graph? The number of cells stays fixed but the input positions vary? If this is the case, please clarify this in the experimental section.


2. In the experimental section it would be good to provide more details regarding the time needed for training the model and the number of parameters of the network (also in comparison with [Li et al 2020 a])


3. The paper should fit within the 9-page limit. I suggest that the background section could be shortened a bit, moving some of its parts to the Appendix.


##########################################################################

Questions during rebuttal period:



Please address and clarify the cons above



#########################################################################

I spotted some typos:

(1) Figure 1(right): reduces -> reduce

(2) Page 5 Decoder section: an temporally -> a temporally

(3) Page 9 Experiments and result section: MP-PDE solvers obtains -> MP-PDE solvers obtain

(4) Page 9 section 4.4:  acccurately -> accurately

(5) Page 10 section 7: many diciplines-> many disciplines

(6) Page 10 section 7: paving a way -> pave the way



**Summary Of The Paper:**

The paper proposes a framework to solve generic PDEs written in conservation form, in an autoregressive fashion. The grid domain is considered as a graph, whose nodes contain information about solutions to the PDE until a given times steps, node positions, time steps, and hyperparameters information about the PDE. The node’s labels are first embedded in a latent space. Then message passing is applied and the output is decoded to predict the next n time steps solutions together.
The network is trained with an additional loss which promotes zero stability of the solver, by constraining the network to be invariant to small perturbations in the input solutions of previous time steps.
The approach is validated on a parameterized family of 1d and 2d PDEs, showing accuracy and generalization results that are above the state of the art of neural solvers and classical solver schemes. In addition, the paper show promising results dealing with irregular domains with different types of boundary conditions.


**Summary Of The Review:**

In my opinion, the paper brings significant advances to the field of neural PDE solvers. The proposed push forward trick, together with the temporal bundling trick, are valuable contributions as they impose favorably condition on autoregressive solvers (zero stability), improve the accuracy of other existing approaches as [Li et al 2020 a] and the inference time of the solver.
The approach is well evaluated experimentally,  showing accuracy and generalization results that are above the state of the art of neural solvers and beat classical solver schemes in inference time.
For these reasons I consider the manuscript to be accepted at ICLR 2022.

---

> ### Author Response · Authors · 2021-11-16
> **Response to Reviewer wxXT**
>
> Thanks for a very detailed review, a detailed feedback and your positive score.
>
> **Generalization to multiple spatial resolutions**: Because we have used an MPNN as the backbone of the solver, the same MP-PDE solver can operate on graphs with widely varying input discretization and resolution, on irregular and regular grids. This is firstly due to the general property of graph neural networks to operate on input graphs of different sizes, and secondly because we use relative positions x_j - x_i as part of the edge features of the network. Thus, we can deploy on different resolutions, with the same setup is what we were referring to in our paper. In all our experiments, we train the same architecture separately on different resolutions (fixed spatial position for every resolution). As discussed below this can, and probably, will be extended in the future. When you talk of generalization, however, we think that you are probably referring to the statistical notion of generalization, which is training on one grid and testing on another of a different resolution. Strictly speaking, this would be out-of-domain generalization, which we do not expect to work properly, unless we include multiple resolutions in our training set. We shall clear up this confusion in the wording of our paper draft.
>
> But since it is an interesting point we have included another appendix (Appendix H) to test out-of-domain generalization for spatial resolution. We found that if we train on fine grids (one fixed resolution), we can generalize reasonably to coarser ones, but not the other way around. The results are pretty good concerning that we really only train on one fixed grid with fixed spatial points. Nonetheless, in a practical setting, we would probably just train on multiple resolutions at once, fostering the generalization capabilities. We have not done this in our experiments, however, plan to do this in the future. So far, we also have only used fixed spatial coordinates for the different resolutions; we want to expand towards different spatial input coordinates and maybe even optimize the integration grid.
>
> A strong motivation for developing our MP-PDE solver is to come up with a framework which can be applied simultaneously to different equations (experiment E2 and E3), different boundary conditions (experiment WE1-WE3), different grids (different resolution, and regular vs irregular), even different spatial dimensions would in principle be possible. A “one solver to rule them all” would however require a huge and diverse training set. We therefore only tested generalization capabilities of MP-PDE solvers by generalizing across subsets of the above mentioned tasks. The point we are trying to make is that our trained solvers work out of the box for many different setting without fiddling around with the inputs, the “how well” needs as usual be determined by the information contained in the training set.
>
> **Time needed for training and number of parameters**: We have extended the implementation and training details in Appendix F. We hope to have provided some interesting insights. Thank you very much for pointing this out!
>
> **9-page limit**: Of course. We should note though that the current manuscript does fit the page limit (ignoring ethics and reproducibility statements)
>
> **Typos**: Thanks for spotting these. Fixed!

---

### Official Review · Reviewer_6VsU · 2021-11-03

**Correctness:** 3
**Technical Novelty And Significance:** 3
**Empirical Novelty And Significance:** 3
**Recommendation:** 6
**Confidence:** 4

**Main Review:**

The ideas of the paper are interesting: the pushforward trick is appealing. The architecture, however, looks like a separate idea, which is not connected to the loss. I.e., the paper proposes two separate innovations, puts one into the title, but experiments confirm the effectiveness of only one. But I like the comparison with the standard numerical solver. However, the study of the architecture lacks the analysis of the systematic convergence of the solution, given by the new architecture, with the number of parameters. Classical solvers do have this property: increase the grid size, get better accuracy. Does the proposed model help with this respect?

Comments:
- There is no ablation study (the loss or the architecture?)
- The dependence of the properties of the method on the number of parameters in the model is not studied (at least, comments are welcome).


**Summary Of The Paper:**

The paper addresses the stability of training neural PDE solvers. It uses the "pushforward trick": the adversarial-style loss is used by perturbing the input at step $k$ by a certain noise (note, that this idea looks similar to denoising autoencoders). Another trick is to predict K steps simulaneously in time. The architecture itself is graph-based: the nodes are distributed through the domain, and it is an encoder-processsor-decoder architecture. This helps to be closer to classical numerical solvers. The experiments show that the pushforward trick helps to increase "survival time" (which is defined as a time when a certain error exceed the threshold) clearly helps.



**Summary Of The Review:**

Reasonable idea, which proposes a loss that improves stability; however, it has two disconnected contributions and is not so easy to read. Some of the important points are not discussed/analyzed experimentally.

---

> ### Author Response · Authors · 2021-11-16
> **Response to Reviewer 6VsU**
>
> Thanks for a positive review. We found your thoughts and comments very helpful.
>
> **Ablation**: Thanks for pointing this out, we definitely agree with this. We originally performed the comparison but didn't place it in the document because of space constraints, we have put it into Appendix G. We compared a simple CNN against our GNN architecture for the same training setup. You have to apply a number of tricks with padding and add extra information into the input to indicate the location of boundaries/equation parameters/etc, to make things comparable.
>
> What is notable is that the CNN has roughly twice the accumulated error compared to the GNN, for the exact same training setup. So indeed the pushforward trick is doing a lot of the heavy work in our results, but so is the architecture. This is to be expected, but the empirical check is useful. We posit this improvement is due to the fact that CNNs can be shown to be a special case of message passing CNNs, which Velickovic refers to as "representational containment".
>
> Of course, on irregular grids you cannot apply the CNN without resampling tricks, but we didn't pursue this route, so there is no comparison there. Most interesting for us is the performance difference between 1D and 2D CNNs on regular grids (1D convolves over space and treats time as a channel, 2D convolves over space and time). It seems that treating time dimension as channel is much more effective than looking at a space-time grid. This was a strong motivation for MP-PDE solvers.
>
> **Parameter and architecture choices**: We have extended the implementation and training details in Appendix F. We hope to have provided some interesting insights. Thank you very much for pointing this out! Appendix G also contains a number of ablations empirically justifying our design choices.
>
> **Convergence analysis**: This is certainly very interesting and something we have discussed many times amongst ourselves. We have not studied the convergence properties of the method with grid spacing and indeed we have no guarantees that we will converge to exact PDE solutions as grid pitch is taken to zero. This is, of course, a general limitation of purely learned methods, such as ours, and it is indeed an interesting area of research to finding hybrid methods, which port over those convergence guarantees from classical solvers while including some learnable components.
>
> One potential direction, which we think would be interesting, was put forward by Bar-Sinai et al. (2019). This uses a neural network to select a finite difference/volume stencils from a family of stencils that are known to have certain order and accuracy constraints. Each stencil approximates derivatives (or fluxes) to a given polynomial order, and the network learns which of the family solves with the best stability and long term accuracy over a rollout. There is a downside of this though, that this assumes smoothness of the solution, which is not true for the Burgers' equation or at boundaries. And digging into their appendix you see that actually they only ever use polynomial accuracy order 1, which we suspect is due to the aforementioned issue. So in general, we would say that this is a very important issue but it is also an open question for the community.
>
> **Ease of reading**:
> We would like to make our paper as readable as possible. Do you have suggestions on how to do this? We are open to improving it.

---

### Official Review · Reviewer_5QFf · 2021-11-04

**Correctness:** 3
**Technical Novelty And Significance:** 2
**Empirical Novelty And Significance:** 2
**Recommendation:** 6
**Confidence:** 3

**Main Review:**

I think the paper has a clarity issues and still in its early stages from that perspective.

In particular, the METHOD section is very vague and not clearly explained. Notation are introduced without definition and no explanation for the background needed to understand the notation. I suggest you lead with an example and motivate from there instead of going the general route.

Also, it is not clear what is the difference between using GNN and other regular networks in solving PDEs--the authors did not mention that in the paper.


**Summary Of The Paper:**

The paper utilizes neural message passing to propose another neural solver for PDEs.


**Summary Of The Review:**

The paper has a clarity issues and many notations are not introduced or defined.

--Edit-- The authors either explained or addressed the issues in the paper with other reviewers. I am happy to recommend the paper for publication.

---

> ### Author Response · Authors · 2021-11-16
> **Response to Reviewer 5QFf**
>
> Thanks for your review.
>
> **Clarity and vagueness**: We would not like the writing to be a hindrance to readers and will make any changes that improve the manuscript. Would you mind providing an example of where we are vague or where clarity is missing? Indeed, your point is somewhat in contradiction to reviewers mh3h and wxXT, who found the paper is clear and well written. Nonetheless, it would certainly be helpful to indicate where you are having difficulties.
>
> **Notation without definitions**: Again, please provide examples and we will accomodate them where possible.
>
> **Lead with an example**: Thanks for this suggestion. Given space constraints we originally chose not to lead with an example, but we could certainly provide one of, say, what the updates would look like for solving the Heat equation for FDM, FVM, and a WENO scheme. This could be added to the background section (section 2).
>
> **GNN vs. other networks**: Thanks for pointing this out, we definitely agree with this. We originally performed the comparison but didn't place it in the document because of space constraints, we have put it into Appendix G. We compared a simple CNN against our GNN architecture for the same training setup. You have to apply a number of tricks with padding and add extra information into the input to indicate the location of boundaries/equation parameters/etc, to make things comparable.
>
> What is notable is that the CNN has roughly twice the accumulated error compared to the GNN, for the exact same training setup. So indeed the pushforward trick is doing a lot of the heavy work in our results, but so is the architecture. This is to be expected, but the empirical check is useful. We posit this improvement is due to the fact that CNNs can be shown to be a special case of message passing CNNs, which Velickovic refers to as "representational containment".
>
> Of course, on irregular grids you cannot apply the CNN without resampling tricks, but we didn't pursue this route, so there is no comparison there. Most interesting for us is the performance difference between 1D and 2D CNNs on regular grids (1D convolves over space and treats time as a channel, 2D convolves over space and time). It seems that treating time dimension as channel is much more effective than looking at a space-time grid. This was a strong motivation for MP-PDE solvers.

---

### Official Review · Reviewer_mh3h · 2021-11-05

**Correctness:** 4
**Technical Novelty And Significance:** 3
**Empirical Novelty And Significance:** 3
**Recommendation:** 8
**Confidence:** 4

**Main Review:**

The paper is very well written with: (1) thorough and detailed literature review; (2) intuitive explanation of motivation; (3) clear methodology and approach used in designing the architecture; (4) extensive details in empirical study and design of experiment. Overall, it is a good paper  with strong motivation, clear objective and solid empirical results. It is worthwhile to mention that authors design the training approach, i.e. pushforward trick, to reduce the instability due to error propagation and allow the network to solve versatile tasks.


**Summary Of The Paper:**

This paper proposes a message passing neural network to solve PDEs. As a graph neural network for solving PDEs, it proposes a few notable features, e.g. embedding in the encoder for equation-wise representation, a difference of inputs in the message function, and so on. The proposed method also adopts notable training methods to reduce error propagation.

**Summary Of The Review:**

Overall, it is a good paper with a complete analysis of the design of architecture, training/testing approach and extensive experiments.

---

> ### Author Response · Authors · 2021-11-16
> **Response to Reviewer mh3h**
>
> We thank reviewer mh3h for a very positive review. We hope that the updates we added to the paper have further strengthened it. Please let us know if you have any queries.

---

### Author Response · Authors · 2021-11-16
**General response**

We thank all the reviewers for the constructive feedback. We were pleased to see that a number of reviewers found the work thorough and clear, that our motivation was strong, that the pushforward trick was novel, and that our empirical results and ablations were significant and insightful.

In response to queries, we have added several experiments and additions to our paper draft. We think that all of them are valuable contributions, and we are very thankful for the suggestions.

The most important changes are:
1) We have extended the implementation and training details in Appendix F for a more complete picture.
2) We have added several architecture ablations in Appendix G:
* We compare the MP-PDE solver against a vanilla CNN on regular grids, demonstrating that the message passing architecture performs better than regular convolution.
* We also compare convolution over time with stacking time in the channel dimension, to justify our design choice of stacking time in the channel dimension.
* We ablate the processor and the decoder part of our architecture an quantify the amount by which they improve our model.
3) We test and investigate the generalization capability of our MP-PDE solvers upon multiple spatial resolutions in Appendix H.

We want to further remark that due to the page limit (9-pages without reproducibility and ethical statement), we are forced to put important aspects of our work to the appendix. Especially the newly added Appendix G and Appendix H we consider very important for our work, again thanks for the valuable inputs. For publication, we plan to move e.g. exemplary 2D results back to the main paper.
We will release code for data generation, for baseline methods, and for our MP-PDE solver upon publication.

---

### Decision · Program_Chairs · 2022-01-20

**Decision:**

Accept (Spotlight)

**Comment:**

This paper proposes a message passing neural network to solve PDEs. The paper has sound motivation, clear methodology, and extensive empirical study. However, on the other hand, some reviewers also raised their concerns, especially regarding the lack of clear notations and sufficient discussions on the difference between the proposed method and previous works. Furthermore, there is no ablation study and the generalization to multiple spatial resolution is not clearly explained. The authors did a very good job  during the rebuttal period: many concerns/doubts/questions from the reviewers were successfully addressed and additional experiments have been performed to support the authors' answers. As a result, several reviewers decided to raise their scores, and the overall assessment on the paper turned to be quite positive.